# LLM-guided Hierarchical Search for End-to-end Reasoning Intensive Retrieval

## Abstract

Search systems are increasingly used for *reasoning-intensive* queries, where what makes a document relevant requires understanding or reasoning over the query–document relation rather than relying on surface vocabulary or topical similarity. The standard recipe – a cheap embedding-based retriever followed by an LLM verifier – works only when the embedding model places the right documents in its top-$k$, an assumption that recent reasoning-intensive IR benchmarks show often fails to hold even for SOTA embedding models (Su et al., 2025; Weller et al., 2025). Many recent works propose query-side fixes such as query rewriting and agentic loops, which have shown promise in bringing LLM reasoning to bear on the search process but can be brittle to the embedding model's effectiveness and to the LLM's ability to rewrite the query from its parametric knowledge. In this paper, we explore a different paradigm – *LLM-guided hierarchical search* – in which an LLM interacts with the corpus directly via a hierarchically navigable search index, with no embedding model in the loop at search time. We propose LATTICE, an instantiation of this paradigm with two technical contributions: (i) a top-down construction of the search index using LLM judgements over multi-level document summaries; and (ii) a robust LLM-guided hierarchical search algorithm that mitigates noisy, context-dependent LLM scores via cross-branch reference nodes and path-aggregated latent scores. Through extensive experiments on the reasoning-intensive BRIGHT benchmark, base LATTICE with an off-the-shelf LLM achieves 46.7 nDCG@10 (matching the best fine-tuned ensemble baseline overall). A lightweight ensemble LATTICE[++] that fuses LATTICE with cheap retrieval reaches **49.1 nDCG@10**. A controlled same-LLM comparison against sliding-window reranking shows that reranking offers a better tradeoff at low LLM token budgets, but after a moderate token budget LATTICE converges to a higher asymptote. We further show that LATTICE works with open-weight LLMs and remains competitive on traditional IR benchmarks.

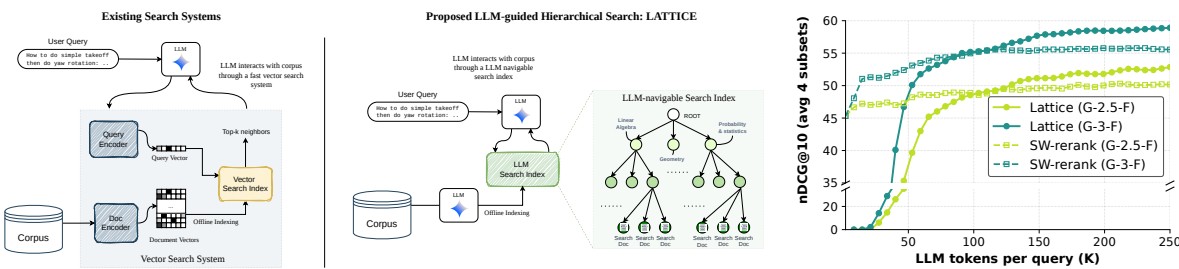

Figure 1: **LATTICE Overview (left).** Existing systems probe the corpus through a vector index built with an embedding model; LATTICE replaces it with an *LLM-navigable search index* constructed offline by an LLM, which walks the index directly at query time – no embedding model in the loop. **LLM-controlled test-time scaling (right).** Average over four BRIGHT subsets, identical LLM and prompt: sliding-window reranking (dashed) over a BGE-Reasoner-Embed top-300 shortlist wins at low budgets by inheriting the shortlist's base nDCG@10; LATTICE (solid) converges to a higher asymptote.

# 1 Introduction

The retrieval workloads that modern search systems are expected to handle are getting harder. A search engine is no longer used only for queries where keyword or topical similarity already pins down the right document. It is increasingly asked to surface evidence for tasks where what makes a document relevant requires reading both the query and the document carefully – brainstorming a niche research idea, doing deep search on an unfamiliar topic (He et al., 2025), locating legal precedent for a specific fact pattern (Pipitone & Alami, 2024), or searching something with under-specified preferences. In recent literature this is typically referred to as the *reasoning-intensive* retrieval setting (Su et al., 2025; Shao et al., 2025).

The standard information retrieval recipe is to combine a cheap embedding-based retriever (for quick lookup) with a stronger LLM *verifier* (a reranker) that re-orders the retriever's top-$k$ (Sun et al., 2023; Reddy et al., 2024). When the embedding model is largely right about the top-$k$, this split is hard to beat: fast, cheap, and the verifier handles the residual ordering noise. But recent reasoning-intensive benchmarks show that the assumption underneath the split breaks down: even state-of-the-art embedding models fail to place the right documents in the top-$k$ when relevance depends on a relation, an argument, or a chain of inference (Su et al., 2025; Shao et al., 2025), and recent theoretical work shows that no fixed-dimensional dense embedding can represent arbitrary top-$k$ ranking relationships (Weller et al., 2025). Anything the retriever misses, the verifier cannot recover. The natural question, then, is how to put an LLM's reasoning capacity to work *inside* the retrieval step, not just on top of it.

Most recent work answers this by pushing the LLM upstream of the retriever rather than into it. The cleanest examples are *query rewriting* – have the LLM produce a hypothetical answer the query might be paired with (HyDE (Gao et al., 2023)), or expand the user's query into a longer, more informative form (Query2Doc (Wang et al., 2023)) – and then run a standard retrieve-then-rerank over the rewritten query. *Agentic retrieval* extends this into a loop: read what was retrieved, decide what is still missing, refine the query, retrieve again, repeat until satisfied (Jin et al., 2025; Zhang et al., 2024). Both approaches work in many settings, but they share a structural property worth noticing: the strong LLM only indirectly interacts with the corpus. Its reasoning shapes the queries that get sent and the candidates that get re-ordered, but the documents that come back are still chosen by the cheap retriever. This makes the pipeline brittle in several characteristic ways. **(a)** The rewriting LLM may simply not know beforehand what the right answer looks like (Mallen et al., 2023); it then probes the corpus with variations of its own parametric beliefs, which an agentic loop reinforces rather than corrects (Xie et al., 2024; Sun et al., 2025). **(b)** Even when the rewrite is good, the embedder may fail to understand a richer, more information-dense query and falls back to surface similarity, surfacing topically related but semantically wrong candidates (Weller et al., 2025). **(c)** Many real corpora already come with structure – section hierarchies, document–passage relations induced by chunking (Karpukhin et al., 2020; Lewis et al., 2020) – which a flat retrieve-then-rerank pipeline discards entirely, forcing the LLM to score passages out of the context they were written in.

In this paper we explore an alternative search paradigm – *LLM-guided hierarchical search* – in which the LLM interacts with the corpus directly, with no embedding model in the loop at search time. In this paradigm, we first offline build an LLM-navigable hierarchical search index over the corpus: a semantic tree whose leaves are documents and whose internal nodes carry compact LLM-written summaries of the documents below them. At query time, a search LLM walks this index with best-first frontier search, scoring children at selected internal node and descending into the most promising subtree (Figure 1). Structurally this resembles HNSW-style hierarchical nearest-neighbour search (Malkov & Yashunin, 2018); the difference is that the relevance judgement at each step is produced by an LLM reading content rather than by a vector dot product against a fixed embedding. The building blocks here are established – semantic hierarchical indexing (Sarthi et al., 2024), LLM-based clustering (Pham et al., 2024), listwise LLM scoring (Sun et al., 2023), and LLM-guided tree search (Yao et al., 2023); our contribution is not any single component, but showing why their naive combination fails, the mechanisms that fix it, and that the result is competitive with the strongest fine-tuned systems.

Although conceptually simple, making this paradigm work pulls in two non-trivial design problems. **(1) Building an index the LLM can reliably navigate.** Bottom-up embed-and-cluster constructions like RAPTOR (Sarthi et al., 2024) give noisy partitions of the corpus – especially at higher levels, where

partitions should reflect broad conceptual distinctions but instead conflate documents that merely share surface vocabulary. We therefore build the hierarchy top-down using LLM judgements over multi-level document summaries, so the same in-context reasoning that makes the LLM a strong ranker organises the corpus it will later navigate. **(2) Robustly navigating it with an off-the-shelf LLM.** An LLM can compare $O(10)$ candidates reliably in a single prompt (Sun et al., 2023; Qin et al., 2023; Pradeep et al., 2023b; Zhuang et al., 2024; Liu et al., 2024), and the scores it returns are noisy and context-dependent – the same node, presented alongside different siblings, can score very differently (Sun et al., 2023; Qin et al., 2023; Tang et al., 2024; Zheng et al., 2023). Naively walking the highest-scoring path at each level commits early and abandons branches that turn out to be correct. We address this with two mechanisms: *cross-branch calibration*, in which each scoring prompt is augmented with reference nodes drawn from other parts of the tree so the LLM has a stable comparison baseline across slates; and *path-aggregated scoring*, in which we fit slate-independent latent scores from the full scoring history and accumulate them along the root-to-leaf path, so the search is guided by cumulative evidence rather than by any single local score.

We validate these design choices on the reasoning-intensive BRIGHT benchmark (Su et al., 2025) and three traditional IR benchmarks NQ, SciFact, SciDocs, spanning corpus sizes from 5K to 2.68M documents (Thakur et al., 2021). On BRIGHT, base LATTICE with a single zero-shot LLM (Gemini-3-Flash) achieves 46.7 nDCG@10 – exceeding the ensembled BGE-Reasoner-0928 (46.4), within 0.1 of DIVER v3 (46.8), and ahead of both on 5 of 12 subsets. A lightweight ensemble LATTICE$^{++}$ that fuses LATTICE with BM25 and a dense retriever reaches **49.1 nDCG@10**. In a same-LLM, same-prompt comparison against a sliding-window reranker over a strong dense shortlist (Figure 1), reranking wins at low token budgets but LATTICE converges to a higher asymptote after a moderate budget. LATTICE also works with open-weight LLMs (Qwen3.5-27B matches Gemini-2.5-Flash on StackExchange average at $\sim 4\times$ lower per-query cost).

Overall, the goal of this paper is to show promise for LLM-guided hierarchical search – in which an LLM interacts with the corpus directly through a navigable search index, rather than probing the corpus through the noisy view of an embedding-based retriever – as a practical alternative to retrieve-then-rerank on reasoning-intensive workloads. LATTICE has $O(\log N)$ asymptotic search complexity, but because each step is an LLM call, in its current form it is best suited to applications that can afford tens of seconds of latency per query – deep research, legal QA, technical QA.

## 2 Methodology

This section describes LATTICE in two parts: an offline procedure that organises a corpus into an LLM-navigable hierarchical search index (Section 2.1), and an online procedure through which a search LLM walks this index to answer a query (Section 2.2).

### 2.1 Offline Hierarchical Search Index Construction

We formalise the index as a rooted tree $T = (V, E, \phi)$. The node set $V = V_L \cup V_I$ consists of leaf nodes $V_L$ – one per document in the corpus $D$ – and internal nodes $V_I$, each representing a cluster of leaf documents below it. Every node carries a textual representation $\phi(v)$: for leaves, the document content; for internal nodes, an LLM-written summary of the documents in its subtree. We write $C(v)$ for the children of $v$ and constrain the branching factor to $|C(v)| \leq M$.

**Bottom-up construction.** To build such an index, we begin from the natural starting point taken by prior work like RAPTOR (Sarthi et al., 2024): a bottom-up construction driven by an embedding model $\mathcal{E}$ and an off-the-shelf clustering routine such as $k$-means. Concretely, starting from $V_0 = V_L$ (one node per document), at each level $\ell = 0, 1, \ldots$, we embed the textual representations $\{\mathcal{E}(\phi(v)) : v \in V_\ell\}$, cluster the resulting vectors into at most $M$ groups, instantiate a new internal node per group whose children are the corresponding $V_\ell$ nodes, and use a summariser LLM to write the new node's $\phi(\cdot)$ from the textual representations of its children. The procedure terminates once $|V_\ell| \leq M$, at which point the surviving nodes become the children of a single root. The full procedure is given as Algorithm 1 in Appendix B.3.

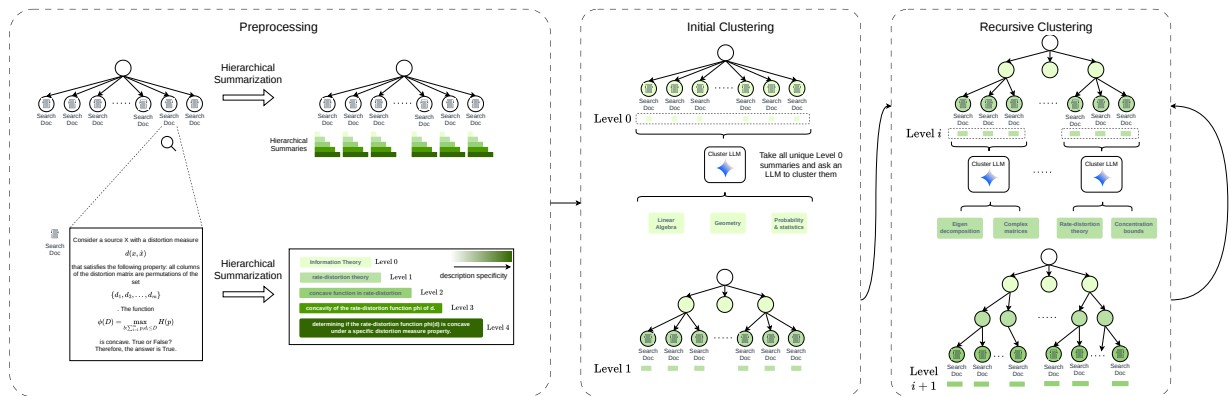

Figure 2: Top-down construction of the LATTICE index. Starting from a root containing all leaf documents, we recursively partition any node with more than $M$ children: an LLM is shown multi-level summaries of the node's leaves and asked to organise them into at most $M$ coherent topics. Each topic becomes a new internal node whose LLM-written description serves as $\phi(v)$; leaves are reassigned per the LLM's mapping.

While conceptually simple and practical to implement, we find that bottom-up indices do not work well at the corpus sizes considered in this work, particularly when passages come from independent sources rather than from a shared parent document; Table 4 reports the corresponding ablation. We hypothesise that this is because a RAPTOR-style index inherits the noise behaviour of the embedding model on which it is built. At the lowest levels of the tree the noise is benign, since surface similarity and conceptual similarity tend to agree at that granularity. At higher levels, however, the partition should reflect broad conceptual distinctions (e.g. separating *population dynamics* from *biochemical signalling* in a biology corpus), and embedding-based clusters tend to mix such conceptually distinct sub-clusters whenever they happen to share surface vocabulary (Weller et al., 2025). The noisy higher-level partitions are then summarised by an LLM, locking the conflation into the cluster summaries the search will later see.

**Top-down construction with an LLM.** To address this, we propose LATTICE's *top-down tree construction* algorithm: a fully LLM-supervised clustering procedure that replaces the embed-and-cluster step at every level with an LLM call. The high-level structure mirrors hierarchical $k$-means: begin with a single root containing every leaf, and recursively partition any cluster whose size exceeds a threshold. The key change is operationalisation – we want the LLM, rather than an embedding clustering routine, to decide how to partition each cluster (Viswanathan et al., 2024; Pham et al., 2024).

The simplest LLM-only instantiation is impractical. Given a node $v$ to partition, we would concatenate the textual representations $\{\phi(v_i) : v_i \in V_L(v)\}$ of all leaves under $v$ and ask an LLM to (i) identify at most $M$ topical clusters with short titles and (ii) map each input to one cluster. Even for a moderately sized cluster ($\sim 10^4$ documents at a few hundred words each), the prompt is already in the millions of tokens, well outside the context limits of any LLM we can afford to call recursively along the tree.

**Hierarchical summarisation pre-processing.** To make the LLM-only construction practical, we preprocess each document with a hierarchical summarisation step (left panel of Figure 2). A lightweight summariser LLM $\mathcal{S}$ produces $k$ summaries of each leaf's textual representation $\phi(v_l)$ at increasing specificity, with the rule that the level-$i$ summary uses at most $2^{i+1}$ words, so the descriptive capacity doubles at every level. We write the resulting summaries as

$$\phi^{(i)}(v_l) = \mathcal{S}\big(\phi(v_l), i\big), \qquad i = 0, 1, \ldots, k-1, \qquad |\phi^{(i)}(v_l)|_{\text{words}} \leq 2^{i+1}.$$

The summaries trade off context length for descriptive specificity: $\phi^{(0)}(v_l)$ is a one-to-two-word topic and $\phi^{(k-1)}(v_l)$ is a one-sentence description (a worked example for a rate-distortion-theory document is shown in Figure 2; the full summariser prompt is in Figure 11, Appendix D). We use $k = 5$ throughout, which is roughly $\log_M N$ deep for the corpora we work with ($N \in [10^4, 10^6]$, $M \in [10, 20]$). For efficiency, $\mathcal{S}$ is realised by a small LLM (Gemini-3.1-Flash-Lite or Qwen3-4B); to amortise the prompt overhead we batch the summarisation, feeding 20 documents in one prompt and asking the model to return hierarchical summaries for all 20 in its output.

**Hierarchical summaries as a dynamic interface to the corpus.** Once the multi-level summaries are available, they serve as a token-efficient, depth-adaptive interface between a cluster LLM CLUSTERLLM and the (sub-)corpus it is asked to partition. To split a cluster node $v$ into smaller clusters, we proceed as follows (Algorithm 3; the middle and right panels of Figure 2 illustrate one application at the root and one recursive application deeper in the tree):

1. **Choose the summary level.** Let $V_L(v)$ be the leaves under $v$. A SELECTSUMMARYLEVEL subroutine picks the most specific level $i^\star$ such that the deduplicated set $U^{(i^\star)}(v) = \{\phi^{(i^\star)}(v_l) : v_l \in V_L(v)\}$ fits within CLUSTERLLM's context budget. This adapts to the depth of the partition: near the root the level is coarse (a few-word topic per document), deeper in the tree it is fine (a sentence per document).

2. **Cluster.** The deduplicated set $U^{(i^\star)}(v)$ is passed to CLUSTERLLM (prompt in Figure 12, Appendix D), which is asked to (i) identify between $M_{\min}$ and $M$ topical clusters along with a short LLM-written title per cluster, and (ii) return a mapping from each input summary to a cluster.

3. **Reassign.** Each topical cluster becomes a new internal node, with the cluster title as its representation $\phi(\cdot)$. Every leaf $v_l \in V_L(v)$ is reassigned to the new internal node corresponding to $\phi^{(i^\star)}(v_l)$ under the mapping.

4. **Recurse.** Any newly created internal node whose child count exceeds $M$ is added to a partitioning queue and processed in the same way. The recursion terminates when every node satisfies $|C(v)| \leq M$.

The depth-adaptive level choice in step 1 means partitions near the root are made over coarse summaries – enough to discriminate broad domains – while partitions deeper in the tree are made over fine summaries that capture finer topical distinctions.

**Hybrid construction for large corpora and corpora with a natural parent–child hierarchy.** In practice, two situations call for a hybrid construction that combines a bottom-up first pass with the proposed top-down algorithm. (a) When the corpus already carries a natural parent–child hierarchy – e.g. chunks derived from longer documents such as Wikipedia articles or research papers – we use that hierarchy directly to form the initial clusters (each parent document becomes a cluster whose leaves are its chunks). (b) When the corpus is too large for the top-down clustering to run directly over all $N$ leaves (as is the case for NQ at 2.6M passages), we first run the bottom-up construction of Algorithm 1 to obtain an initial layer of coarse clusters, and treat those as pseudo-leaves. In both cases, the top-down algorithm above is then applied with the resulting pseudo-leaves in place of $V_L$. This is more efficient as the LLM-driven top-down clustering operates on a much smaller leaf set – and, in case (a), it also preserves the natural organisation of the corpus inside the index, so that the search LLM can later evaluate individual passages within the context of their parent document.

## 2.2 Online LLM-guided Hierarchical Search

At query time, the index from Section 2.1 is walked by an LLM-guided best-first frontier search. At each search step we use the LLM as an in-context *listwise scorer*: given a query $q$ and the textual descriptions of a small set of candidate nodes, it returns a graded relevance score in $[0, 1]$ per candidate. Formally, the search LLM is a function

$$\mathcal{L}(q, [\phi(v_1), \ldots, \phi(v_n)]) = [s_1, \ldots, s_n], \qquad s_i \in [0, 1]. \tag{1}$$

The prompt has four blocks – instruction, query, candidate descriptions, and a JSON output schema requesting per-candidate reasoning and a score in $[0, 1]$; the full template is in Figure 10, Appendix D. Listwise scoring lets the LLM compare candidates in context within a single call and reduces the per-expansion budget from $n$ pointwise calls to one.

It is useful to read each per-child score $s_i$ as the LLM's estimate of the *value* of expanding the subtree rooted at $v_i$ – loosely, how relevant the best leaf reachable under $v_i$ is to the query. The search then behaves like a best-first heuristic search whose value function is an LLM reading content (Hart et al., 1968; Russell & Norvig, 2021); this LLM-as-value-function framing has also seen recent use in LLM tree-search for reasoning (Yao et al., 2023; Hao et al., 2023). The non-trivial part is converting these per-step value estimates into a globally

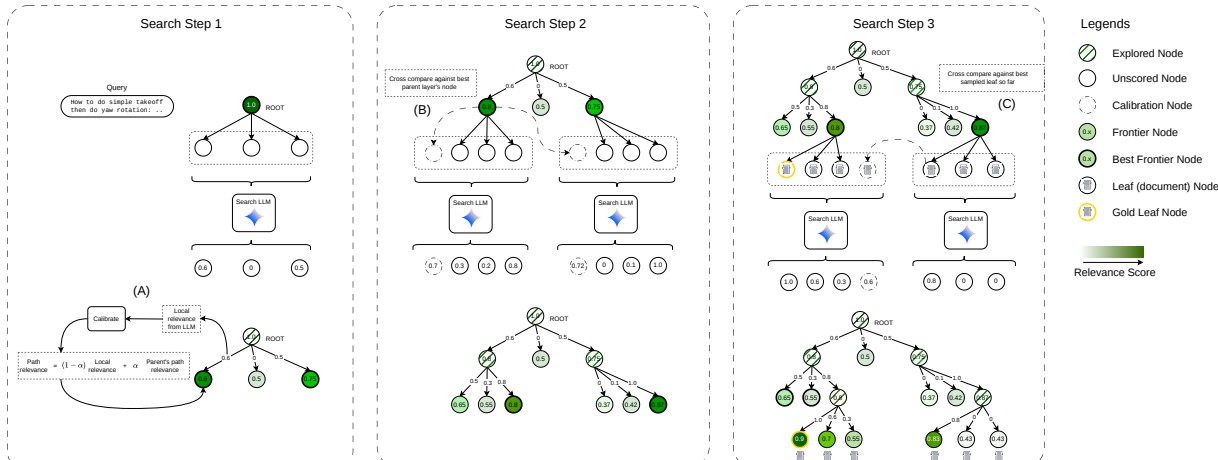

Figure 3: LATTICE tree search. A frontier $F$ of unexpanded internal nodes is ordered by path-relevance. At each iteration the top-$B$ frontier nodes are expanded: the LLM scores a slate of each node's children plus cross-branch reference nodes; raw scores are converted to slate-independent latent scores by fitting a per-slate-bias linear model over the running history; latent scores are accumulated along the root-to-leaf path into a global path-relevance score that drives the next iteration's frontier priority.

consistent ranking over the tree. A naive instantiation – score each frontier node's children with $\mathcal{L}$, push them onto the frontier by raw score, repeat – fails for two reasons:

- **Slate dependence.** A listwise scorer's output depends on which other candidates appear in the prompt (Sun et al., 2023; Qin et al., 2023). A node $v$ scored alongside a strong sibling can receive a lower number than the same $v$ scored alongside a weak one; reading raw scores as direct estimates of $v$'s relevance conflates the LLM's view of $v$ with the company $v$ happened to keep.

- **Locality.** Even a perfectly comparable raw score reflects a comparison among only the $\leq M$ candidates in a single slate, which is too narrow a signal to globally order the thousands of paths the search is choosing between.

Our expansion routine addresses both: a *latent-score estimator* strips the slate-dependent bias from the raw $s_v^i$, and a *path-relevance* aggregator carries information along the search trajectory. The rest of this section describes the search loop and one full expansion in execution order; the full pseudocode is in Algorithm 4 (Appendix B).

**Search loop.** The search (Figure 3; pseudocode in Algorithm 4, Appendix B) maintains

- a max-priority queue $F$ of unexpanded internal nodes, keyed by a path-relevance score $\hat{p}_{\mathrm{rel}}(v)$ defined below;

- a running prediction set Pred of leaves already scored;

- a score history of (slate id, $v$, $s_v^i$) triples accumulated from every listwise call.

We seed $F$ with the root and set $\hat{p}_{\mathrm{rel}}(v_{\mathrm{root}}) = 1$. Each iteration pops the top-$B$ entries of $F$ and expands them in parallel as described in the next four paragraphs.

**Slate construction.** For each expanded node $v$, the LLM is shown a *slate* – the list of candidates to score in a single listwise call. The slate is $v$'s children $C(v)$ together with a small calibration set $\mathrm{Aug}(v)$ drawn from elsewhere in the tree. The calibration set has a single purpose: it gives the LLM reference candidates that recur across slates, anchoring the resulting scores to a strong comparison. The composition of $\mathrm{Aug}(v)$ depends on the type of children $v$ has:

- **When $C(v)$ are internal nodes:** $\mathrm{Aug}(v)$ contains the highest-scoring sibling of $v$ from a previously expanded slate – a cross-branch anchor the LLM has already evaluated.

- **When $C(v)$ are leaves:** Aug($v$) contains $\ell$ top-scoring leaves drawn from Pred, sampled with probability proportional to their current latent score – the strongest leaves seen so far, used as anchors.

The full slate is passed to $\mathcal{L}$ (Eq. 1), which returns raw scores $s_{v'}^i \in [0,1]$ for every item in slate $i$. The resulting triples $(i, v', s_{v'}^i)$ are appended to the score history.

**Latent score estimation.** Calibration anchors do not by themselves eliminate slate dependence – a node's raw score still varies with the particular composition of its slate. We estimate a slate-independent *latent score* $\hat{s}_v$ per node by modelling each observed raw score as a linear function of the latent score, plus a slate-specific shift – in the same spirit as recent work that calibrates noisy LLM-judge scores by fitting a latent-skill model over many comparisons (Li et al., 2024a; Zheng et al., 2023):

$$s_v^i \approx a \cdot \hat{s}_v + b^i, \tag{2}$$

where $a$ is a global scale shared across slates and $b^i$ is a per-slate bias. After every expansion, all parameters are re-estimated jointly by minimising mean-squared error over the full score history,

$$\min_{a, \{\hat{s}_v\}, \{b^i\}} \sum_i \sum_{v' \in \text{slate}^i} \left(s_{v'}^i - (a \cdot \hat{s}_{v'} + b^i)\right)^2. \tag{3}$$

A node that appears across multiple slates has its $\hat{s}_v$ jointly constrained by all of them. The per-slate $b^i$ term is what distinguishes the estimator from a naive per-node average over raw scores: averaging conflates a node's intrinsic relevance with the strength of the company it happened to keep, biasing estimates downward for nodes that tended to appear alongside stronger candidates and upward for those that tended to appear alongside weaker ones.

Eq. 2 is identifiable only up to an affine reparameterisation of $(a, \{\hat{s}_v\}, \{b^i\})$. We pin it down with a small $\ell_2$ penalty on all parameters in Eq. 3 (selecting the minimum-norm solution, equivalent to fixing $a=1$ and centering $\{b^i\}$), and map the fitted scores back onto the $[0, 1]$ scale consumed by the search via max-normalisation, $\hat{s}_v \leftarrow a\hat{s}_v / \max_u a\hat{s}_u$ – the same scale as the raw scores and the root's path relevance (Eq. 4 below). A node observed in only a single slate remains identified because the calibration anchors, which recur across slates, pin that slate's bias $b^i$. Full estimator details are in Appendix B.1.

**Path-relevance update.** Latent scores are still local – each $\hat{s}_v$ reflects a comparison among the $O(M)$ candidates in $v$'s slate, a narrow signal for a search over $\Theta(N)$ documents. To carry information along the search trajectory, we accumulate latent scores along the root-to-node path into a *path-relevance* score

$$\hat{p}_{\text{rel}}(v) = \alpha \cdot \hat{p}_{\text{rel}}(\text{parent}(v)) + (1 - \alpha) \cdot \hat{s}_v, \tag{4}$$

with $\hat{p}_{\text{rel}}(v_{\text{root}}) = 1$ and $\alpha \in [0, 1]$ controlling the trade-off between ancestral and local evidence. This is an exponential moving average up the path; with $\alpha = 0.5$ (our default) it weights deeper (more recent) splits more heavily, which we want because they are made over finer-grained summaries that more directly reflect the leaves below.

**Frontier and prediction-set update.** After expanding $v$, each newly scored internal child $v' \in C(v)$ is pushed onto $F$ with priority $\hat{p}_{\text{rel}}(v')$, and each leaf child is added to Pred. The next iteration pops the top-$B$ entries of $F$ and repeats. The search terminates after $N$ iterations and returns the top-$K$ entries of Pred ranked by $\hat{p}_{\text{rel}}$, for a per-query cost of $B \cdot N$ listwise LLM calls and asymptotic $O(\log_M N)$ tree depth.

## 3 Experiments

This section evaluates LATTICE along three axes: ranking quality on the reasoning-intensive BRIGHT benchmark (Section 3.2); scalability to traditional IR corpora up to 2.68M documents (Section 3.3); and per-component contribution via ablations (Section 3.4).

Table 1: nDCG@10 performance of various retrievers and rankers on the BRIGHT benchmark. **Bold** represents overall best numbers, underline represents best numbers among non-ensemble methods, * denotes subsets with dynamic corpus.

| Method | Ensem-ble | StackExchange | | | | | | | | Coding | | | Theorem-based | | | | Avg. |
|---|---|---|---|---|---|---|---|---|---|---|---|---|---|---|---|---|---|
| | | Avg. | Bio. | Earth. | Econ. | Psy. | Rob. | Stack. | Sus. | Avg. | Leet.* | Pony | Avg. | AoPS* | ThQ.* | ThT. | |
| *Retrieve-then-rerank* | | | | | | | | | | | | | | | | | |
| Rank-R1-14B | ✗ | 26.6 | 31.2 | 38.5 | 21.2 | 26.4 | 22.6 | 18.9 | 27.5 | 14.7 | 9.2 | 20.2 | 10.3 | 9.7 | 11.9 | 9.2 | 20.5 |
| Qwen1.5-7B+InteRank-3B | ✗ | 31.9 | 51.2 | 51.4 | 22.4 | 31.9 | 17.3 | 26.6 | 22.4 | 23.8 | 24.5 | 23.1 | 19.4 | 13.5 | 19.3 | 25.5 | 27.4 |
| GPT-4 + Rank1-32B | ✗ | 32.0 | 49.7 | 35.8 | 22.0 | 37.5 | 22.5 | 21.7 | 35.0 | 25.7 | 18.8 | 32.5 | 25.8 | 10.8 | 22.9 | 43.7 | 29.4 |
| ReasonIR | ✗ | 41.7 | 59.8 | 53.2 | 32.0 | 43.6 | 28.8 | 38.7 | 36.0 | 34.0 | 33.2 | 34.8 | 29.4 | 7.9 | 32.6 | 47.7 | 37.3 |
| ReasonRank | ✗ | 46.8 | 62.7 | 55.5 | 36.7 | 54.6 | 35.7 | 38.0 | 44.8 | 27.5 | 29.5 | 25.6 | 35.5 | 14.4 | 42.0 | 50.1 | 40.8 |
| XRR2 (GPT-4+BM25→G-2.5-F) | ✗ | 47.4 | 63.1 | 58.2 | 38.5 | 52.9 | 37.1 | 37.6 | 44.6 | 28.4 | 21.9 | 35.0 | 31.8 | 15.7 | 34.4 | 45.5 | 40.3 |
| *Iterative-retrieve-then-rerank* | | | | | | | | | | | | | | | | | |
| BGE-Reasoner (no ens.) | ✗ | 51.1 | 64.3 | 62.6 | 45.3 | 52.7 | 44.3 | 41.5 | 46.9 | 22.3 | 14.4 | 30.2 | 34.5 | 14.2 | 41.7 | 47.5 | 42.2 |
| DIVER v2 | ✓ | 52.2 | 68.0 | 62.5 | 42.0 | 58.2 | 41.5 | 44.3 | 49.2 | 33.8 | 34.8 | 32.9 | 38.6 | 19.1 | 44.3 | 52.6 | 45.7 |
| BGE-Reasoner-0928 | ✓ | 52.0 | 68.5 | 66.4 | 40.6 | 53.1 | 43.2 | 44.1 | 47.8 | 35.3 | 29.0 | 41.6 | 40.7 | 17.2 | 46.5 | 58.4 | 46.4 |
| DIVER v3 | ✓ | 51.8 | 66.0 | 63.7 | 42.4 | 55.0 | 40.6 | 44.7 | 50.4 | 39.9 | 32.5 | 47.3 | 39.7 | 17.2 | 46.4 | 55.6 | 46.8 |
| *LLM-guided Hierarchical Search* | | | | | | | | | | | | | | | | | |
| LATTICE (G-2.5-f) | ✗ | 52.0 | 66.3 | 63.0 | 47.4 | 54.0 | 47.6 | 37.6 | 48.2 | 26.9 | 19.9 | 34.0 | 32.6 | 12.0 | 38.0 | 47.9 | 43.0 |
| LATTICE (G-3.0-f) | ✗ | 55.8 | 69.4 | 65.0 | 48.8 | 61.0 | 53.9 | 42.2 | 50.4 | 33.0 | 22.9 | 43.2 | 34.6 | 13.7 | 44.0 | 46.0 | 46.7 |
| LATTICE++ (G-2.5-f) | ✓ | 53.6 | 71.4 | 66.9 | 46.5 | 55.3 | 45.4 | 43.7 | 45.9 | 36.7 | 28.5 | 44.8 | 36.7 | 14.2 | 43.6 | 52.4 | 46.5 |
| LATTICE++ (G-3.0-f) | ✓ | 55.7 | 73.2 | 67.1 | 45.3 | 58.3 | 52.4 | 45.5 | 48.2 | 41.9 | 32.6 | 51.2 | 38.3 | 16.2 | 46.9 | 51.9 | 49.1 |

## 3.1 Experimental Setup

**Benchmark and metrics.** BRIGHT (Su et al., 2025) contains 12 reasoning-intensive retrieval tasks (corpus sizes 7.9K–414K) where queries require multi-step inference rather than keyword match. Three subsets (marked * in Table 1) use a *query-dependent dynamic corpus* – up to 10K documents are excluded per query at eval time – which we discuss in Section 3.2. We report nDCG@10 for ranking quality and Recall@100 for coverage. Per-dataset statistics are in Appendix B.2.

**Baselines.** We compare against the strongest published systems on BRIGHT: ReasonIR (Shao et al., 2025), DIVER v1/v2/v3 (Long et al., 2025), ReasonRank (Liu et al., 2025), BGE-Reasoner (Chen et al., 2025), and XRR2. The current leaders – DIVER and BGE-Reasoner – are fine-tuned ensembles combining four retrieval sources (dense retriever, BM25, pointwise reranker, listwise reranker) (Long et al., 2025; Chen et al., 2025).

**Implementation.** We use Gemini-2.5-Flash as the summariser LLM and the CLUSTERLLM for offline index construction. For the main comparison we test two search LLMs – Gemini-2.5-Flash and the stronger Gemini-3-Flash – to measure search-LLM-capability effects. Ablations use Gemini-3-Flash unless stated otherwise. Default hyperparameters: $N=20$ iterations, beam $B=2$, path-relevance momentum $\alpha=0.5$, $\ell=10$ leaf calibration nodes, branching factor $M=10\text{–}20$. LATTICE operates directly on the *raw* user query throughout – no query rewriting is used at any stage of the hierarchical search; rewritten queries appear only in the cheap components of LATTICE++ (below) and in baselines. Index construction for a 100K-document corpus (e.g. Biology) takes $\approx 6$ hours; more details in Appendix B.

**LATTICE++ ensemble.** For a better comparison against the fine-tuned ensemble leaders on the BRIGHT leaderboard – which combine multiple retrieval sources – we report a lightweight ensemble LATTICE++ that augments LATTICE with two cheap external signals. For each query we take per-document scores from three sources (LATTICE, BM25, and the BGE-Reasoner-Embed dense retriever), min-max normalise each source's scores to $[0, 1]$, and form the per-document fused score as a weighted sum with weights $w_{\text{LATTICE}}=0.6$, $w_{\text{BM25}}=0.2$, $w_{\text{BGE-RE}}=0.2$. BM25 and BGE-Reasoner-Embed are both run on the BGE-reasoner rewritten query. A per-component decomposition of the ensemble is in Appendix B.5.

## 3.2 Main Results on BRIGHT

**vs. state-of-the-art systems.** With Gemini-3-Flash as the search LLM, base LATTICE reaches 46.7 nDCG@10 overall – exceeding the four-component fine-tuned ensemble BGE-Reasoner-0928 (46.4) by 0.3 and trailing DIVER v3 (46.8) by 0.1, while remaining single-LLM, zero-shot, and fine-tuning-free. With the weaker

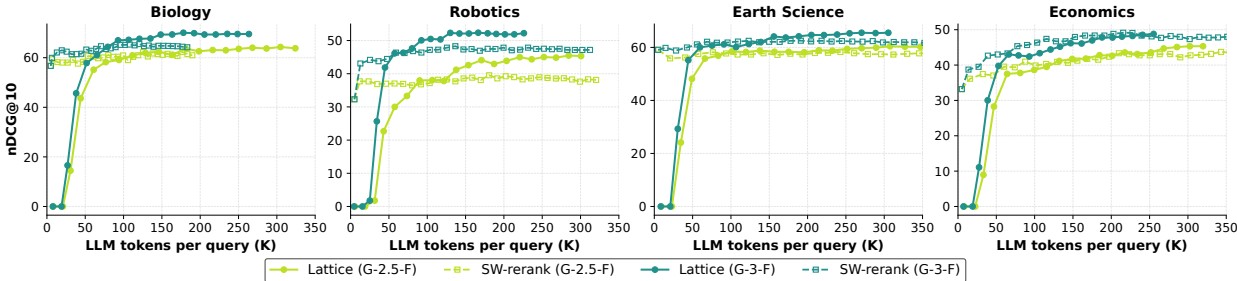

Figure 4: Same-LLM test-time scaling on four BRIGHT subsets. Solid = LATTICE; dashed = sliding-window reranker over a BGE-Reasoner-Embed top-300 shortlist; yellow = Gemini-2.5-Flash, teal = Gemini-3-Flash.

Gemini-2.5-Flash, LATTICE reaches 43.0, behind the best ensembles by 3–4 points. Among non-ensemble baselines, LATTICE with Gemini-3-Flash beats the strongest (BGE-Reasoner without ensembling, 42.2) by +4.5 points overall and the next-best zero-shot baseline XRR2 (40.3) by +6.4, with a +8.4 gap on StackExchange average (55.8 vs. 47.4). The low-cost ensemble LATTICE[++] (Section 3.1) pushes this further: with Gemini-3-Flash it reaches **49.1 nDCG@10** overall, ahead of DIVER v3 (46.8) and BGE-Reasoner-0928 (46.4) by 2.3–2.7 points, and setting a new best on 8 of 16 columns in Table 1. We emphasise that 49.1 is an ensemble number – it fuses LATTICE with BM25 and BGE-Reasoner-Embed scores – whereas the base hierarchical-search method alone reaches 46.7; a per-component decomposition is in Appendix B.5. The gain over base LATTICE is largest on the dynamic-corpus subsets where the static index suffers from stale summaries (Section 3.2).

**Controlled comparison vs. retrieve-then-rerank.** The cleanest measure of the algorithmic gap between LATTICE and retrieve-then-rerank is a same-LLM, same-prompt comparison. We use a sliding-window reranker (SW-rerank) that takes the BGE-Reasoner-Embed top-300 shortlist over the BGE-Reasoner rewritten query and reranks it with the *same* search LLM and the *same* scoring prompt as LATTICE; only the search algorithm differs. Following standard practice, SW-rerank uses a window size of 20 with a top-10 carry-over between successive windows. Figure 4 reports per-subset token–accuracy curves on Biology, Robotics, Earth Science, and Economics. SW-rerank wins at low token budgets because the shortlist itself already has non-trivial nDCG@10 for free, but it plateaus once its retrieval ceiling is hit; LATTICE overshoots SW-rerank after a moderate budget and converges to a higher asymptote on Biology (+5.3 with Gemini-3-Flash), Robotics (+5.0), and Earth Science (+3.8), and matches it on Economics (+0.7). The crossover holds for both Gemini-2.5-Flash and Gemini-3-Flash, with the gap widening on the stronger LLM.

Table 2: nDCG@10 with open-source and proprietary search LLMs on five BRIGHT subsets (zero-shot, $B$=2, $N$=20, $\ell$=10). **Cost per query** is computed from input/output tokens measured directly from LATTICE runs and per-token rates from Google AI Studio (Gemini) and DeepInfra (Qwen); full pricing and token counts are listed in Appendix B.7. Qwen and Gemini-3.1-Flash-Lite run in non-thinking mode; Gemini-2.5-Flash and Gemini-3-Flash use thinking.

| Model | Bio | Ear | Eco | Psy | Rob | Avg | Cost / query |
|---|---|---|---|---|---|---|---|
| Qwen3.5-0.8B | 2.9 | 2.5 | 0.3 | 1.5 | 0.9 | 1.6 | $0.003 |
| Qwen3.5-2B | 16.0 | 10.0 | 13.5 | 9.6 | 4.3 | 10.7 | $0.006 |
| Qwen3.5-4B | 53.3 | 51.2 | 38.9 | 42.7 | 31.8 | 43.6 | $0.009 |
| Qwen3.5-9B | 52.4 | 54.7 | 40.6 | 49.8 | 34.5 | 46.4 | $0.010 |
| Qwen3.5-27B | 66.9 | 63.3 | 46.8 | 59.4 | 47.1 | 56.7 | $0.11 |
| Gemini-2.5-Flash | 66.3 | 63.0 | 47.4 | 54.0 | 47.6 | 55.7 | $0.44 |
| Gemini-3.1-Flash-Lite | 66.8 | 64.2 | 43.6 | 57.6 | 48.9 | 56.2 | $0.06 |
| Gemini-3-Flash | **69.4** | **65.0** | **48.8** | **61.0** | **53.9** | 59.6 | $0.38 |

**Search LLM and dollar cost.** Table 2 reports nDCG@10 on the five StackExchange subsets across the Qwen3.5 open-weight family (0.8B–27B, non-thinking) and three Gemini models, alongside an estimated dollar cost per query (input/output token rates from Google AI Studio and DeepInfra; token counts measured

directly from LATTICE runs). Two findings stand out. First, scaling is the dominant lever: quality climbs steeply from Qwen3.5-2B (10.7) to Qwen3.5-4B (43.6) and saturates around Qwen3.5-27B (56.7), which slightly exceeds Gemini-2.5-Flash (55.7) at ∼4× lower cost ($0.11 vs. $0.44 per query). Second, on the proprietary side, Gemini-3.1-Flash-Lite hits the best accuracy/cost trade-off: 56.2 nDCG@10 at $0.06/query – comparable to Qwen3.5-27B at half the cost and within 2.6 points of Gemini-3-Flash (58.8 at $0.38). Gemini-3-Flash remains the accuracy ceiling and is what we use for the main BRIGHT results. For context, accounting the ensemble leaders' published pipelines at the same open-weight rates gives ≈$0.06/query for DIVER v3 (14B+32B reranking over a top-100 shortlist) and ≈$0.71/query for the full BGE-Reasoner-0928 ensemble (≈3,500 reranker calls across six retriever×size combinations); details in Appendix B.7. LATTICE is thus cost-competitive with the systems it matches on quality, though at higher per-query latency.

**Dynamic-corpus subsets.** On the three *-marked subsets (LeetCode, AoPS, TheoremQA-Q), where >10K documents are excluded per query, LATTICE trails the best ensembles by 9–13 points. Our tree summaries are computed once over the full corpus, so subtrees near excluded leaves become stale and mislead the search. This is atypical of production retrieval, where corpora rarely change per-query, but it points at a real limitation of any static hierarchical index. LATTICE$^{++}$ partly closes the gap on these subsets by leaning on the BM25/BGE side of the ensemble for excluded-aware retrieval. A qualitative failure case is in Appendix C.2; Appendix B.9 shows that a search-guided document-insertion mechanism can adapt the index without a full rebuild (+2.3 nDCG@10 over a static tree on TheoremQA-T).

### 3.3 Generalisation Beyond Reasoning-Intensive Retrieval

To test whether LLM-guided hierarchical search remains competitive beyond reasoning-intensive retrieval, we evaluate LATTICE on three traditional BEIR (Thakur et al., 2021) datasets (Table 3): SciFact (5K), SciDocs (25K), and NQ (**2.68M**). LATTICE is competitive on all three – 75.8 on SciFact (above both BGE-large and BGE-reranker), 22.4 on SciDocs (between the two), and 66.8 on NQ (well above

Table 3: nDCG@10 on BEIR datasets. LATTICE scales to a 2.68M-document corpus on NQ.

| Dataset | Corpus | BGE-large | BGE-rer. | LATTICE |
|---------|--------|-----------|----------|---------|
| SciFact | 5K | 74.6 | 74.1 | **75.8** |
| SciDocs | 25K | **22.6** | 17.0 | 22.4 |
| NQ | 2.68M | 55.0 | **69.0** | 66.8 |

BGE-large 55.0 and within 2.2 of BGE-reranker 69.0) – and the NQ result additionally shows the approach scales to million-document corpora. That said, on these workloads LATTICE does not add enough value over standard retrieve-then-rerank to justify its substantially higher per-query cost; the regime where it is worth the compute is reasoning-intensive retrieval, where the cheap retriever structurally fails.

### 3.4 Analysis and Ablations

Unless stated otherwise, ablations use Gemini-3-Flash as the search LLM and report nDCG@10 on Biology (or StackExchange average where indicated). Together, the ablations attribute LATTICE's gains to its three ingredients: the *index* (Table 4: fixed traversal, swapped construction), the *search algorithm* (Figure 5, Table 5: calibration, path aggregation, beam width, reasoning on a fixed index), and the *LLM budget* (Figure 4).

**Tree construction.** Table 4 compares bottom-up RAPTOR-style construction (embed + spectral cluster) against our top-down LLM-guided construction under identical structural constraints ($M$, depth). Top-down wins on both evaluated datasets, with a 2-point gain on Biology and a much larger 12-point gain on TheoremQA-Theorems. We attribute the TheoremQA gap to embedding-based clustering's well-known weakness on dense mathematical

Table 4: Top-down LLM-guided construction vs. bottom-up RAPTOR-style embedding clustering.

| | Biology | | TheoremQA-T | |
|--------------|---------|-------|-------------|-------|
| **Construction** | nDCG@10 | R@100 | nDCG@10 | R@100 |
| Bottom-Up | 67.6 | 89.5 | 33.5 | 64.6 |
| Top-Down | **69.4** | **91.3** | **46.0** | **76.2** |

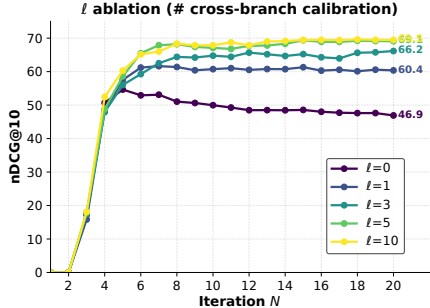 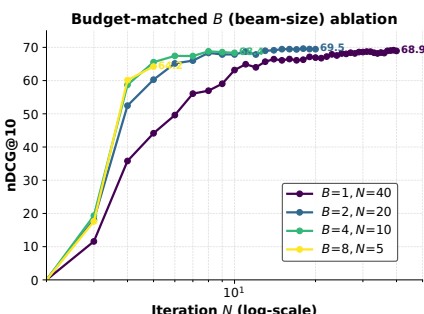

Figure 5: Hyperparameter ablations on Biology (Gemini-3-Flash). (left) Cross-branch calibration is essential. (right) Under a fixed budget $B \times N{=}40$, depth beats breadth. Trends match on Gemini-2.5-Flash (Appendix B.6).

notation: theorems that share surface symbols but address unrelated concepts cluster together, producing incoherent subtrees the search LLM cannot navigate.

**Cross-branch calibration.**    Figure 5 (left) sweeps the number of cross-branch calibration nodes $\ell$. Without calibration ($\ell{=}0$), Gemini-3-Flash plateaus at 46.9 nDCG@10 – a 22.6-point gap from the $\ell{=}10$ asymptote of 69.5. Performance improves monotonically and saturates around $\ell{=}5$–10 (69.1, 69.5), supporting that slate-dependent score bias is intrinsic to listwise prompting.

**Beam size vs. iterations.**    Figure 5 (right) fixes the search budget at $B \times N{=}40$ and varies the split. Depth dominates breadth: $B{=}1, N{=}40$ and $B{=}2, N{=}20$ reach 68.9 and 69.5; $B{=}4, N{=}10$ drops to 68.4; $B{=}8, N{=}5$ collapses to 64.2. The ordering is identical on Gemini-2.5-Flash (Appendix B.6). This justifies our default $B{=}2$: it preserves intra-iteration parallelism while giving the search enough rounds to commit to a path, refit calibration, and re-rank.

**Traversal components.**    Table 5 ablates each traversal component on StackExchange average, using Gemini-2.5-Flash because the Gemini-3-Flash API does not expose a way to disable internal reasoning, needed for the "no reasoning" row (`thinking_budget=0`). All three components contribute meaningfully; path-relevance aggregation is the largest single contributor (+3.4 nDCG@10 vs. $\alpha{=}0$), followed by explicit reasoning (+2.7 vs. `thinking_budget=0`) and score calibration

Table 5: Ablation on LATTICE's core traversal components (StackExchange avg., nDCG@10).

| Configuration | Avg. |
|---|---|
| **LATTICE (Full)** | **52.00** |
| – No Score Calibration | 49.56 |
| – No Path Relevance ($\alpha{=}0$) | 48.62 |
| – No Reasoning | 49.33 |

(+2.4 vs. raw slate scores). Each component addresses a distinct failure mode: path-relevance supplies a global signal beyond individual slate scores, reasoning lets the LLM evaluate options carefully, and calibration separates node-intrinsic relevance from slate-specific bias so cross-branch comparisons remain meaningful.

# 4   Related Work

**LLMs in Retrieve-then-Rerank Pipelines.**    The dominant IR paradigm pairs a cheap retriever with an LLM reranker (pointwise or listwise) (Reddy et al., 2024; Sun et al., 2023; Zhu et al., 2023), with performance bottlenecked by the initial retrieval stage (Rathee et al., 2025b). LLMs are also increasingly used as backbones for dense embedding models (Luo et al., 2024; Lee et al., 2025).

**Generative and Long-Context Retrieval.**    Alternative paradigms include Generative Retrieval (e.g. DSI (Tay et al., 2022; Li et al., 2024b)), which maps queries directly to document IDs but struggles to scale and update (Pradeep et al., 2023a), and Long-Context Retrieval, which loads the entire corpus into the LLM's context (Lee et al., 2024a; Gupta et al., 2025) but is computationally infeasible at moderate scale. LATTICE offers a middle ground via a semantic hierarchy the LLM navigates efficiently.

**Hierarchical Indices.** Hierarchical structures are long-standing for efficient search in large output spaces – hierarchical softmax (Morin & Bengio, 2005), tree-based extreme classification (Prabhu & Varma, 2014; Yu et al., 2022; Gupta et al., 2022), and HNSW for ANN (Malkov & Yashunin, 2018) – but those hierarchies are geometric, not semantic, and traversal is a vector dot-product. Closer to our setting, RAPTOR (Sarthi et al., 2024) builds a semantic hierarchy by bottom-up clustering and summarisation, and EHI (Kumar et al., 2024) learns a hierarchical index jointly with a dense encoder; both still traverse via embedding similarity. LATTICE instead uses an LLM as an *active traversal agent* that reads node content and reasons at each step.

**Adaptive Corpus-Graph Retrieval.** GAR (MacAvaney et al., 2022), QUAM (Rathee et al., 2025a), and SlideGAR (Rathee et al., 2025b) alternate between reranking an initial pool and expanding graph neighbours over a precomputed corpus graph; RGS (Xu & Chen, 2025) greedily searches an ANN proximity graph. These are the flat-graph counterparts of our hierarchical traversal; LATTICE's search structure is an LLM-built *semantic* hierarchy rather than an embedding-built proximity graph that inherits the limitations of the underlying embeddings.

**LLM-based Clustering.** A related line uses LLMs to organise documents into coherent clusters or topics. TopicGPT (Pham et al., 2024) prompts an LLM to discover topics top-down, and Viswanathan et al. (2024) show that LLMs cluster more reliably than embedding $k$-means on semantically nuanced corpora. Our top-down construction (Section 2.1) extends this into a recursive hierarchical setting.

**Reasoning as Pre-processing and Agentic IR.** Query expansion (Wang et al., 2023; Gao et al., 2023) brings LLM reasoning to bear before retrieval but leaves the core search mechanism unchanged (Long et al., 2025; Shao et al., 2025). Agentic IR (Jin et al., 2025; Zhang et al., 2024; He et al., 2025) treats retrieval as a multi-step loop in which the LLM probes an opaque search tool with generated queries, which is brittle when the right answer is not anticipated in the LLM's parametric knowledge. Graph-RAG (Edge et al., 2024; Zhang et al., 2025) uses LLMs to reason over pre-structured knowledge graphs but with a limited retrieval role. LATTICE differs in that the LLM *is* the search mechanism: the semantic tree constrains the agent's action space to make global search tractable, while the agent's reasoning drives traversal – close in spirit to LLM-guided tree search for reasoning (Yao et al., 2023; Hao et al., 2023).

## 5 Conclusion

The standard retrieve-then-rerank pipeline is bottlenecked by the cheap retriever, acutely so on reasoning-intensive workloads where it cannot place the right documents in its top-$k$; query rewriting and agentic loops push the LLM upstream, but it still sees the corpus only through the same noisy retriever. LATTICE brings the LLM *inside* the retrieval step: it walks an LLM-navigable semantic hierarchy directly, with no embedding model in the loop at search time, via a top-down LLM-guided index construction and a calibrated, path-aggregated traversal that mitigates the slate-dependence and locality of raw LLM scores. With a single zero-shot LLM, LATTICE matches the best four-component fine-tuned ensembles on BRIGHT; a three-source ensemble LATTICE$^{++}$ sets a new state-of-the-art at 49.1 nDCG@10; and the approach runs on open-weight backbones at $\sim 4\times$ lower cost. Since each search step is an LLM call, LATTICE best suits quality-over-latency applications; next steps include search-guided index updates and distilling the traversal into smaller models.

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

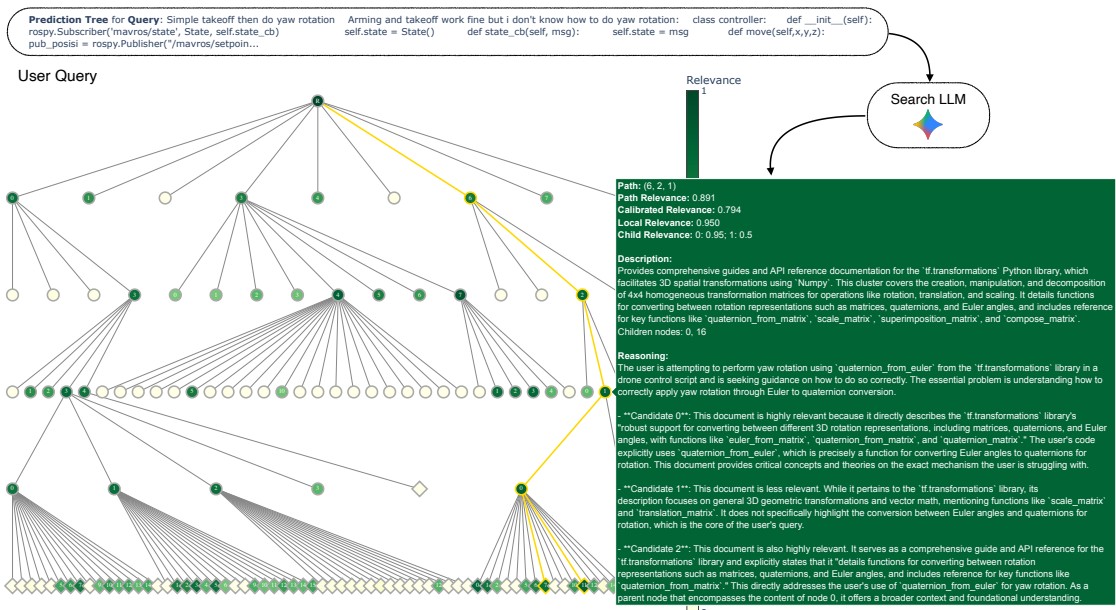

Figure 6: An illustration of the search process of LATTICE for a real query from the BRIGHT benchmark. The color of each node corresponds to its computed path relevance; the highlighted yellow path shows the path to ground-truth documents. The search LLM makes a step-by-step decision at each internal node to determine which branch to explore next. The expanded callout provides a "glass box" view into one such decision, detailing the LLM's explicit reasoning process as it scores the child nodes.

# A    Limitations and Future Work

Our work introduces a promising framework for hierarchical search, but it also presents several avenues for future research. One of the limitation of our current approach is the use of a **static semantic tree**. Future work could explore methods for efficient, localized updates to the tree's summaries, allowing the hierarchy to adapt to a changing corpus without the need for a full reconstruction.

Second, the index imposes a **hard partition**: each document lies on a single root-to-leaf path. Multi-topic or bridging documents may therefore be reachable only through one of their topics. Beam search, cross-branch calibration, and path aggregation mitigate wrong-branch entries at search time – and coverage remains strong in practice (Recall@100 of 74.8% on the StackExchange average, ahead of fine-tuned flat retrievers; Appendix B.8) – but construction-time topical ambiguity is not explicitly modelled. Nothing in the traversal requires a partition, however: a document could be replicated as a leaf under multiple internal nodes and deduplicated in the prediction set, and extending the CLUSTERLLM to emit soft or multi-cluster assignments is a natural extension we have not evaluated. Relatedly, the **branching-factor constraint** $|C(v)| \leq M$ means skewed topic distributions are absorbed as extra depth – dense topics become deeper subtrees, as in hierarchical $k$-means – rather than forced merges; this preserves navigability but increases traversal cost for queries into dense regions, and makes the fidelity of intermediate nodes dependent on the quality of the LLM's recursive splits.

Third, the **offline tree construction process**, while a one-time cost, can be computationally intensive for extremely large corpora due to the repeated use of LLMs for clustering and summarization. Research into more efficient construction methods, perhaps by combining traditional clustering for the lower levels with LLM-based summarization for only the top, most abstract layers, could further improve scalability.

While our greedy, best-first traversal is effective in a zero-shot setting, the entire process could be framed as a reinforcement learning problem, where the search LLM is an agent trained to optimize a policy for navigating the tree to maximize retrieval rewards. We believe that exploring these directions will further establish hierarchical, LLM-driven navigation as a powerful approach for reasoning-intensive information retrieval.

# B    Implementation Details

## B.1    Hyperparameters

This section provides a detailed list of all hyperparameters and implementation choices used in our experiments to ensure full reproducibility.

### B.1.1    Offline Tree Construction

- **Maximum Branching Factor ($M$):** We set the maximum number of children for any node to $M = 10 - 20$.

- **Embedding Model ($\mathcal{E}$):** For bottom-up tree implementation we use `gecko` (Lee et al., 2024b) embeddings to generate vector representations for the clustering steps.

- **Clustering Algorithm ($\mathcal{C}$):** For bottom-up tree implementation uses an iterative spectral clustering (Ng et al., 2001) algorithm to partition nodes into at most $M$ clusters at each level of the hierarchy.

- **Summarization LLM:** We use `Gemini-2.5-flash` for all summarization tasks (both for internal nodes in the bottom-up method and for the multi-level document summaries in the top-down method). The exact prompt template used is detailed in Appendix D.

- **Top-Down Summary Levels:** For the top-down method, we generate 5 levels of hierarchical summaries for each document.

**StackExchange Tree Construction.**    For datasets where documents are passages from a smaller set of source articles (the StackExchange sub-datasets in BRIGHT), we leverage this inherent structure. We form initial clusters by grouping all passages belonging to the same source document. If any resulting cluster contains more than $M$ passages, we further subdivide it by grouping passages based on their location proximity within the source document, continuing until all sub-clusters satisfy the branching factor constraint. This approach produces more coherent initial clusters than embedding-based grouping alone, since passages from the same source document are often semantically related.

### B.1.2    Online Traversal

- **Search LLM ($\mathcal{L}$):** We use `Gemini-2.5-flash` as the search agent that performs the listwise scoring. The prompt structure is provided in Appendix D.

- **Number of Iterations ($N$):** We run the search for $N = 20$ iterations for all main experiments.

- **Beam Size ($B$):** We use a beam size of $B = 2$ for parallel node expansion in each iteration.

- **Path Relevance Momentum ($\alpha$):** The smoothing factor for the path relevance score is set to $\alpha = 0.5$.

- **Calibration Nodes ($l$):** We augment each leaf slate with $\ell = 10$ cross-branch leaf nodes for calibration, based on our ablation study.

- **Reasoning Budget:** The default "thinking budget" for the LLM's reasoning step is set to $-1$, meaning the model gets to decide how long it wants to think.

- **Latent-Score Estimator:** The latent scores are re-estimated after each batch of slate evaluations by minimizing the objective of Eq. 3 over the full score history with the Adam optimizer (learning rate $10^{-2}$, 100 steps, small weight decay). Eq. 2 is identifiable only up to an affine reparameterisation: for any $c \neq 0$ and $d$, the transformation $(a, \hat{s}_v, b^i) \to (a/c, c\hat{s}_v + d, b^i - (a/c)\,d)$ leaves all predictions unchanged. All parameters are initialized at neutral values ($a{=}1$, $b^i{=}0$, $\hat{s}_v{\approx}0$), and the weight-decay (ridge) term – strictly convex along the degenerate directions – selects the minimum-norm solution

in the equivalence class, playing the same role as fixing $a=1$ or centering $\{b^i\}$. Note that the affine ambiguity is shared across all nodes and therefore cannot change their induced ranking; the only downstream quantity sensitive to absolute scale is the path-relevance mixing of Eq. 4, which the max-normalisation $\hat{s}_v \leftarrow a\hat{s}_v / \max_u a\hat{s}_u$ pins by anchoring the top-scoring node to 1 – the same scale as the raw scores and the root's path relevance. Each squared-error term is additionally weighted by $w(s) = \max\big(0.1, \sigma(s - 0.5)\big)$, mildly emphasizing candidates the LLM scored higher, which we found improves robustness to noise on low-scored candidates.

**Usage of LLMs**  During the preparation of this manuscript, LLM were used as a collaborative writing assistant to aid with drafting, refining prose for clarity and conciseness, and structuring arguments; all core ideas, experiments, and analyses were conducted by the authors.

## B.2  Dataset Details

All experiments are conducted on the BRIGHT benchmark (Su et al., 2025), a comprehensive collection of 12 datasets designed to evaluate reasoning-intensive retrieval. A summary of the statistics for each subset is provided in Table 6.

Table 6: Statistics for the 12 subsets of the BRIGHT benchmark used in our experiments.

| Dataset Subset | # Queries | Corpus Size ($\mathcal{D}$) | Avg. Doc Length |
|---|---|---|---|
| *StackExchange* | | | |
| Biology | 103 | 57,359 | 83.6 |
| Earth Science | 116 | 121,249 | 132.6 |
| Economics | 103 | 50,220 | 120.2 |
| Psychology | 101 | 52,835 | 118.2 |
| Robotics | 101 | 61,961 | 121.0 |
| Stack Overflow | 117 | 107,081 | 704.7 |
| Sustainable Living | 108 | 60,792 | 107.9 |
| *Coding* | | | |
| LeetCode | 142 | 413,932 | 482.6 |
| Pony | 112 | 7,894 | 98.3 |
| *Math* | | | |
| AoPS | 111 | 188,002 | 250.5 |
| TheoremQA-Q | 194 | 188,002 | 250.5 |
| TheoremQA-T | 76 | 23,839 | 354.8 |

The datasets exhibit two key characteristics relevant to our work. First, the StackExchange subsets are composed of passages derived from longer source documents. We leverage this structure for our metadata-based initial clustering in the bottom-up tree construction method. Second, the Coding and Theorem-based datasets (excluding Pony and TheoremQA Theorems) utilize a **query-dependent corpus**, where a unique list of documents (often >10k) must be excluded from the search space for each query. This feature, discussed in our main results analysis, poses a unique challenge for static index structures like our semantic tree.

## B.3  Tree Construction

### B.3.1  Bottom-up

The bottom-up tree construction algorithms are defined in Algorithm 1 and Algorithm 2.

### B.3.2  Top-down

The top-down tree construction algorithm is presented in Algorithm 3 (Section 2.1). Here we detail the two subroutines it relies on.

The **SelectSummaryLevel** function implements a heuristic to find the optimal summary granularity for a given set of leaf nodes. It begins with the most abstract summary level ($i = 1$) and iteratively checks the

---

**Algorithm 1** Bottom-Up Tree Construction

---

1: **Parameters:** Corpus $D$, $\mathcal{E}$, $\mathcal{C}$, Summarize LLM, $M$, Optional InitialClusters
2: **Initialize:** $V_L \leftarrow \{\text{Node}(d) \mid d \in D\}$, $V \leftarrow V_L$, $E \leftarrow \emptyset$
3: **if** InitialClusters is provided **then**
4:      $V_{\text{current}} \leftarrow \text{CreateNodesFromClusters}(V_L, \text{InitialClusters}, V, E)$
5: **else**
6:      Embeddings $\leftarrow \{\mathcal{E}(\phi(v)) : v \in V_L\}$
7:      Clusters $\leftarrow \mathcal{C}(\text{Embeddings})$
8:      $V_{\text{current}} \leftarrow \text{CreateNodesFromClusters}(V_L, \text{Clusters}, V, E)$
9: **end if**
10: **while** $|V_{\text{current}}| > M$ **do**
11:      {Summarize the current layer before clustering}
12:      **for all** $v$ in $V_{\text{current}}$ **do**
13:          $\phi(v) \leftarrow \text{Summarize}(\{\phi(c) \mid c \in C(v)\})$
14:      **end for**
15:      $V_{\text{next\_layer}} \leftarrow \emptyset$
16:      Embeddings $\leftarrow \{\mathcal{E}(\phi(v)) : v \in V_{\text{current}}\}$
17:      Clusters $\leftarrow \mathcal{C}(\text{Embeddings})$
18:      $V_{\text{next\_layer}} \leftarrow \text{CreateNodesFromClusters}(V_{\text{current}}, \text{Clusters}, V, E)$
19:      $V_{\text{current}} \leftarrow V_{\text{next\_layer}}$
20: **end while**
21: $v_{root} \leftarrow \text{NewInternalNode}()$, $\phi(v_{root}) \leftarrow$ ""
22: $C(v_{root}) \leftarrow V_{\text{current}}$
23: $V \leftarrow V \cup \{v_{root}\}$, $E \leftarrow E \cup \{(v_{root}, c) \mid c \in C(v_{root})\}$
24: **return** Tree $T = (V, E)$

---

**Algorithm 2** CreateNodesFromClusters Subroutine

---

1: **function** CreateNodesFromClusters($V_{\text{source}}$, Clusters, $V$, $E$)
2: **Input:**
3:      $V_{\text{source}}$: The set of nodes in the layer to be clustered.
4:      Clusters: The partition of $V_{\text{source}}$'s embeddings from $\mathcal{C}$.
5:      $V, E$: The global node and edge sets for the tree (passed by reference).
6: **Initialize:** $V_{\text{new\_layer}} \leftarrow \emptyset$
7: **for all** cluster $K$ in Clusters **do**
8:      $v_{new} \leftarrow \text{NewInternalNode}()$
9:      $C(v_{new}) \leftarrow \{v \in V_{\text{source}} \mid v \in K\}$
10:      $V \leftarrow V \cup \{v_{new}\}$
11:      $E \leftarrow E \cup \{(v_{new}, c) \mid c \in C(v_{new})\}$
12:      $V_{\text{new\_layer}} \leftarrow V_{\text{new\_layer}} \cup \{v_{new}\}$
13: **end for**
14: **return** $V_{\text{new\_layer}}$

---

number of unique summaries, selecting the first level $i$ where the count of unique summaries is sufficient for meaningful clustering (e.g., greater than $M$) while remaining under a maximum token limit for the LLM context.

The **ClusterLLM** function is realized via a structured prompt (see Figure 12). The LLM is provided with the list of unique summaries and tasked with grouping them into $M$ coherent conceptual clusters. The prompt instructs the model to first generate a short, descriptive title for each of the $M$ clusters, and then to output a mapping from each input summary to one of these cluster titles. The final output is a structured object containing the $M$ topic descriptions (which become the $\phi(v)$ for the new nodes) and the mapping.

---

**Algorithm 3** Top-Down Divisive Tree Construction

---

1: **Parameters:** Corpus $D$, Summarize LLM, Cluster LLM, Max branching factor $M$
2: **Initialize:**
3: For each document $d_l \in D$, generate multi-level summaries $\{\phi(v_l)^i\}_{i=1}^{5}$.
4: $V_L \leftarrow \{\text{Node}(d) \mid d \in D\}$, $V \leftarrow V_L$
5: $v_{root} \leftarrow \text{NewInternalNode}()$, $C(v_{root}) \leftarrow V_L$
6: $V \leftarrow V \cup \{v_{root}\}$, $E \leftarrow \{(v_{root}, c) \mid c \in V_L\}$
7: PartitionQueue $\leftarrow$ new Queue()
8: **if** $|V_L| > M$ **then**
9:     PartitionQueue.enqueue($v_{root}$)
10: **end if**
11: **while** PartitionQueue is not empty **do**
12:     $v \leftarrow$ PartitionQueue.dequeue()
13:     LeafDescendants $\leftarrow$ GetLeafDescendants($v, T$)
14:     $i \leftarrow$ SelectSummaryLevel(LeafDescendants)
15:     UniqueSummaries $\leftarrow$ unique($\{\phi(c)^i \mid c \in \text{LeafDescendants}\}$)
16:     TopicDescs, Mapping $\leftarrow$ ClusterLLM(UniqueSummaries, $M$)
17:     NewChildren $\leftarrow \emptyset$
18:     **for** $j = 1$ **to** $M$ **do**
19:         $v_j' \leftarrow \text{NewInternalNode}()$, $\phi(v_j') \leftarrow \text{TopicDescs}[j]$
20:         $V \leftarrow V \cup \{v_j'\}$, NewChildren $\leftarrow$ NewChildren $\cup \{v_j'\}$
21:     **end for**
22:     ReassignChildren(LeafDescendants, Mapping, NewChildren, T)
23:     $E \leftarrow E \setminus \{(v, c) \mid c \in C(v)\}$ {Disconnect old children}
24:     $C(v) \leftarrow$ NewChildren
25:     $E \leftarrow E \cup \{(v, c) \mid c \in \text{NewChildren}\}$ {Connect new children}
26:     **for all** $v_j'$ in NewChildren **do**
27:         **if** $|C(v_j')| > M$ **then**
28:             PartitionQueue.enqueue($v_j'$)
29:         **end if**
30:     **end for**
31: **end while**
32: **return** Tree $T = (V, E)$

---

### B.4 Tree Traversal

The full pseudocode for the online LATTICE tree search described in Section 2.2 is given in Algorithm 4.

### B.5 LATTICE$^{++}$: Per-Component Ensemble Analysis

The main paper reports LATTICE$^{++}$ (Section 3.2), a low-cost ensemble that augments LATTICE with BM25 and the BGE-Reasoner-Embed dense retriever. For each query, we min-max normalise the per-document scores from each system to $[0, 1]$ and take a weighted sum with weights 0.6 (LATTICE), 0.2 (BM25), and 0.2

---

**Algorithm 4** LLM-guided Hierarchical Search

---

1: **Parameters:** $q, T, \mathcal{L}, B, N, K, \alpha$
2: **Initialize:**
3: Frontier $F \leftarrow$ new MaxPriorityQueue(), Pred $\leftarrow \emptyset$
4: ScoreHistory $\leftarrow \emptyset$, LatentScores $\leftarrow \emptyset$
5: $\hat{p}_{rel}(v_{root}) \leftarrow 1.0$, $F$.push($v_{root}, \hat{p}_{rel}(v_{root})$)
6: **for** $i = 1$ **to** $N$ **do**
7:     Beam $\leftarrow$ Extract top $B$ nodes from $F$
8:     **for all** $v$ in Beam **do**
9:         Slate $\leftarrow C(v) + Aug(v)$
10:         LocalScores $[s_{v'}]_{v' \in \text{Slate}} \leftarrow \mathcal{L}(q, [\phi(v')]_{v' \in \text{Slate}})$
11:         Add $\{(\text{slate\_id}_i, v', s_{v'}) \mid v' \in \text{Slate}\}$ to ScoreHistory
12:     **end for**
13:     LatentScores $\leftarrow$ SolveMLE(ScoreHistory) {Minimize MSE to find all $\hat{s}_v$}
14:     **for all** $v$ in Beam that were just expanded **do**
15:         **for all** $v'$ in Slate **do**
16:             $\hat{s}_{v'} \leftarrow$ LatentScores$[v']$
17:             $\hat{p}_{rel}(v') \leftarrow \alpha \cdot \hat{p}_{rel}(\text{parent}(v')) + (1 - \alpha) \cdot \hat{s}_{v'}$
18:         **end for**
19:         **for all** $v'$ in $C(v)$ **do**
20:             **if** $v'$ is a leaf node **then**
21:                 Add $v'$ to Pred
22:             **else**
23:                 $F$.push($v', \hat{p}_{rel}(v')$)
24:             **end if**
25:         **end for**
26:     **end for**
27: **end for**
28: **return** Top-$K$ nodes from Pred sorted by $\hat{p}_{rel}$

---

(BGE-Reasoner-Embed). BM25 and BGE-Reasoner-Embed are both run over the BGE-Reasoner rewritten queries, following the standard evaluation protocol on the benchmark.

Table 7: Per-component decomposition of LATTICE$^{++}$ (nDCG@10); fusion scheme described in Section 3.1. LATTICE uses Gemini-2.5-Flash; BM25 and BGE-Reasoner-Embed run on the BGE-Reasoner rewritten queries. * denotes subsets with a dynamic corpus.

| Configuration | SE Avg. | Bio. | Earth. | Econ. | Psy. | Rob. | Stack. | Sus. | Cod. Avg. | Leet.* | Pony | Thm Avg. | AoPS* | ThQ.* | ThT. | Avg. |
|---|---|---|---|---|---|---|---|---|---|---|---|---|---|---|---|---|
| BM25 (rewritten query) | 36.4 | 58.3 | 55.2 | 23.4 | 38.3 | 24.8 | 26.7 | 27.9 | 25.3 | 23.8 | 26.8 | 20.1 | 5.6 | 29.5 | 25.1 | 30.5 |
| BGE-Reasoner-Embed (rewritten query) | 43.4 | 56.7 | 59.1 | 33.2 | 45.7 | 32.3 | 39.3 | 37.4 | 34.0 | 26.8 | 41.3 | 37.2 | 15.6 | 45.0 | 51.1 | 40.3 |
| LATTICE (Gemini-2.5-Flash) | 52.0 | 66.3 | 63.0 | 47.4 | 54.0 | 47.6 | 37.6 | 48.2 | 26.9 | 19.9 | 34.0 | 32.6 | 12.0 | 38.0 | 47.9 | 43.0 |
| LATTICE$^{++}$ (Gemini-2.5-Flash) | **53.6** | **71.4** | **66.9** | 46.5 | **55.3** | 45.4 | **43.7** | 45.9 | **36.7** | **28.5** | **44.8** | **36.7** | **14.2** | **43.6** | **52.4** | **46.5** |

Table 7 decomposes LATTICE$^{++}$ (Gemini-2.5-Flash) into its three components. LATTICE alone is the strongest single retriever on average (43.0 nDCG@10), more than 2 points ahead of BGE-Reasoner-Embed (40.3) and 12 points ahead of BM25 over the reasoning-rewritten query (30.5). The ensemble closes a further 3.5-point gap, reaching 46.5. The gains over single LATTICE are concentrated in two places: (i) the dynamic-corpus subsets (LeetCode $19.9 \rightarrow 28.5$, AoPS $12.0 \rightarrow 14.2$, TheoremQA-Q $38.0 \rightarrow 43.6$, TheoremQA-T $47.9 \rightarrow 52.4$), where stale tree summaries hurt the static LATTICE index and the two retrievers contribute orthogonal signal, and (ii) the StackExchange subsets dominated by document-level topic match (Biology $66.3 \rightarrow 71.4$, Earth Science $63.0 \rightarrow 66.9$), where the dense retriever is already strong and the ensemble accumulates a few additional correct retrievals on top of LATTICE's high baseline. On subsets where LATTICE is already well above both retrievers – Economics (47.4 vs. 33.2/23.4), Robotics (47.6 vs. 32.3/24.8), Sustainable Living (48.2 vs. 37.4/27.9) – ensembling provides little additional gain.

## B.6   Hyperparameter Ablations with Gemini-2.5-Flash

For completeness, Figure 7 reports the same hyperparameter sweeps as Figure 5 (Section 3.4) but with Gemini-2.5-Flash as the search LLM. Both trends carry over: nDCG@10 improves monotonically with $\ell$ (calibration is essential) and depth beats breadth under a fixed budget. The absolute scores are 4–10 points lower than with Gemini-3-Flash, but the relative ordering of configurations is identical.

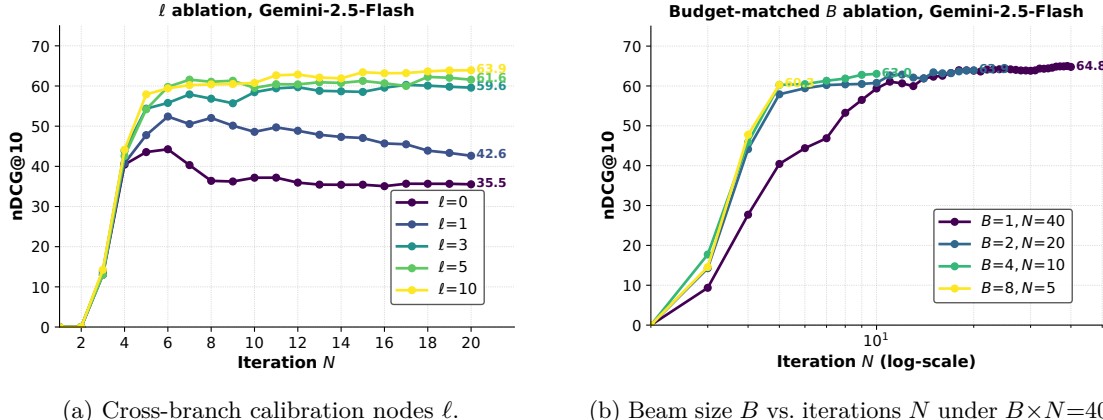

(a) Cross-branch calibration nodes $\ell$.

(b) Beam size $B$ vs. iterations $N$ under $B \times N = 40$.

Figure 7: Hyperparameter ablations on Biology with Gemini-2.5-Flash. Trends mirror the Gemini-3-Flash version in Figure 5 – calibration is essential and depth beats breadth – at uniformly lower absolute nDCG@10.

## B.7   Cost Per Query Breakdown for Table 2

The cost-per-query column of Table 2 is computed as $\bar{T}_{\text{in}} \cdot p_{\text{in}} + \bar{T}_{\text{out}} \cdot p_{\text{out}}$, where $\bar{T}_{\text{in}}, \bar{T}_{\text{out}}$ are the average input and output token counts per query measured directly from LATTICE runs and $p_{\text{in}}, p_{\text{out}}$ are per-token rates from the providers. Per-token rates (\$/1M, in/out): Google AI Studio – Gemini-2.5-Flash 0.30/2.50, Gemini-3-Flash 0.50/3.00, Gemini-3.1-Flash-Lite 0.25/1.50; DeepInfra (Qwen3.5) – 0.8B 0.01/0.05, 2B 0.02/0.10, 4B 0.03/0.15, 9B 0.04/0.15, 27B 0.26/2.40. Measured per-query token counts: Qwen (non-thinking, family avg) $\bar{T}_{\text{in}} \approx 149K$, $\bar{T}_{\text{out}} \approx 27K$; Gemini-2.5-Flash 168K / 155K; Gemini-3-Flash 164K / 98K; Gemini-3.1-Flash-Lite 153K / 14K.

**Ensemble baseline cost accounting.**   The per-query cost estimates for the ensemble baselines quoted in Section 3.2 are obtained by accounting each system's published pipeline at the DeepInfra open-weight rates above. DIVER v3 retrieves with BM25 + a dense retriever and reranks the top-100 shortlist with 14B pointwise and 32B listwise rerankers over multiple passes, totalling ≈243K input and ≈44K output tokens per query, i.e. ≈\$0.06/query. The full BGE-Reasoner-0928 ensemble runs 8B, 14B, and 32B rerankers across six retriever×model-size combinations (≈3,500 reranker calls per query), totalling ≈3.0M input and ≈0.9M output tokens per query, i.e. ≈\$0.71/query. These are estimates from published pipeline descriptions and public rates rather than measurements of the authors' serving stacks; retrieval (BM25/dense) and query-rewriting costs, which are small relative to reranking, are excluded for all systems. Note also that LATTICE's offline index construction is a one-time cost (≈6 hours for a 100K-document corpus, Section 3.1) amortized over all queries, and incremental corpus updates do not require a rebuild (Appendix B.9).

## B.8   Retrieval Coverage on StackExchange

Beyond top-10 ranking, LATTICE surfaces a substantially better candidate set than the strongest retrieval-only baselines. Figure 8 reports Recall@100 on the seven StackExchange subsets of BRIGHT: LATTICE (Gemini-2.5-Flash) achieves 74.8% average recall – +9.5pp over BM25 over the GPT-4 rewritten query, and +4.0pp over the fine-tuned ReasonIR-8B. The largest coverage gains are on the multi-step-reasoning subsets, consistent with the ranking pattern reported in the main paper: when the relevance signal requires reasoning, walking the tree exposes documents that flat first-stage retrievers miss entirely.

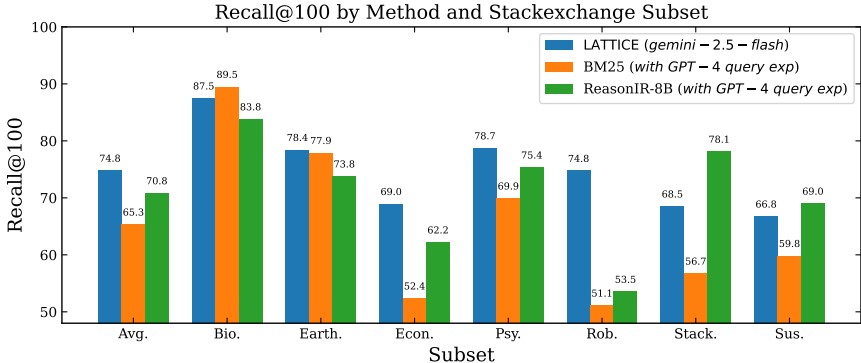

Figure 8: Recall@100 on the seven StackExchange subsets of BRIGHT. LATTICE achieves 74.8% average recall, outperforming BM25 by +9.5 pp and the fine-tuned ReasonIR-8B by +4.0 pp.

Table 8: Dynamic insertion on TheoremQA-T. Documents placed via search-guided insertion outperform a static tree built with the full corpus.

| Method | nDCG@10 | R@100 |
|---|---|---|
| Static Construction (full corpus) | 47.4 | 73.9 |
| Dynamic Insertion (search-guided) | **49.7** | **81.1** |

### B.9 Search-and-Insert Index Updates

A practical concern for any hierarchical index is adaptability to corpus changes: each insertion or deletion can in principle invalidate cluster summaries along the affected root-to-leaf path. We explore a lightweight *search-and-insert* update mechanism that side-steps a full rebuild. Given a new document, we run the online search of Section 2.2 on the existing tree using the document text itself as the query, and attach the document as a leaf of the highest-scoring leaf cluster returned by the search. The cluster's $\phi(\cdot)$ summary is left unchanged.

We evaluate this mechanism on TheoremQA-Theorems by withholding all ground-truth documents during tree construction, building a tree over the remaining corpus, and then inserting the withheld documents via the search-guided procedure above. Table 8 compares retrieval quality against the static baseline – the same tree built once over the full corpus, including the gold documents.

The dynamically updated tree (49.7 nDCG@10, 81.1 Recall@100) *outperforms* the static tree (47.4 nDCG@10, 73.9 Recall@100). Although this result is on a single dataset and we do not claim it as a general phenomenon, it is consistent with a familiar intuition from HNSW-style graph construction (Malkov & Yashunin, 2018): when document placement is made by the same procedure that later retrieves the document, the resulting index is structurally well-aligned with the search algorithm that will traverse it.

## C   Subjective Analysis

### C.1   Sample scoring response from LLM

To provide a more intuitive understanding of our method, Figure 6 presents a qualitative case study of the search process for a real query from the BRIGHT benchmark. The user query is a code snippet asking about "yaw rotation," a complex 3D graphics problem. The figure visualizes the semantic tree and the traversal path taken by LATTICE (highlighted in yellow) to successfully locate a relevant document deep within the hierarchy.

The expanded callout provides a "glass box" view into the search LLM's reasoning at a critical decision point. The LLM's generated **Reasoning** explicitly connects the user's query to the node's topic, noting that the user is "attempting to perform yaw rotation using quaternion_from_euler." It then performs a

Figure 9: Search failing due to dynamically excluded search corpus, red edges denote excluded leaf nodes, gold edges denote ground-truth path

detailed, comparative evaluation of the children nodes. It correctly identifies Candidate 1 as highly relevant because it discusses "support for converting between different 3D rotation representations, including matrices, quaternions, and Euler angles," which directly addresses the user's problem. This example demonstrates that our method does not rely on shallow semantic similarity; instead, the search is an active process guided by the LLM's deep, step-by-step reasoning about the query in the context of the corpus hierarchy.

## C.2 Search failure on dynamic corpus

Figure 9 provides a qualitative case study of a search failure, visually demonstrating the primary challenge our method faces on datasets with a dynamic corpus. The figure shows the search tree for a random query from the AoPS dataset. Red edges indicate leaf nodes that were dynamically excluded for this specific query, while the yellow path highlights the ideal traversal route to the ground-truth document.

As the figure shows, the search agent correctly follows the ground-truth path for the first two levels. However, it then reaches an internal node whose pre-computed summary is now misleading; the summary was generated based on all of its children, including the large number that have since been pruned from the search space (the red nodes). This inaccurate, stale summary causes the search LLM to make an incorrect judgment, deviating from the correct path and ultimately failing to retrieve the relevant document. This example visually confirms the specific failure mode of a static hierarchical index when faced with a dynamic corpus, reinforcing the quantitative analysis in our main results section.

## C.3 A Coarse Error Taxonomy

Our quantitative results permit a coarse attribution of LATTICE's residual errors to pipeline stages. **(1) Coverage vs. ranking.** Recall@100 on the StackExchange subsets averages 74.8% (Appendix B.8):

roughly a quarter of gold documents are never surfaced into the prediction set – failures attributable to index construction and traversal – while the remaining nDCG@10 gap comes from surfaced-but-misranked documents, attributable to leaf scoring and calibration. **(2) Construction errors** are isolated by the matched-traversal ablation of Table 4: replacing the top-down index with a bottom-up one (identical traversal) costs 1.8 nDCG@10 points on Biology and 12.5 on TheoremQA-T, indicating that construction quality is corpus-dependent and dominates on notation-heavy corpora. **(3) Traversal and scoring errors** on static corpora are isolated by the component ablations: removing cross-branch calibration costs 22.6 points on Biology (Figure 5) and 2.4 on the StackExchange average, removing path aggregation 3.4 points, and removing explicit reasoning 2.7 points (Table 5). **(4) Staleness errors** dominate on the dynamic-corpus subsets, where the 9–13 point gap to ensemble baselines (Section 3.2) and the qualitative trace above (Figure 9) attribute failures to summaries computed over since-excluded documents. In short: on static corpora, calibration and construction quality are the components most responsible for residual errors; on dynamic corpora, summary staleness dominates. A per-query automated error-attribution pipeline is an interesting direction for future work.

# D  Prompts

```
You are an intelligent search agent navigating a hierarchical semantic tree of topics. Your mission is to predict the most promising candidates to find
the answer to the user's query using the relevance definition below.

**Relevance Definition:** {relevance_defintion}

---

## USER QUERY

{query}

---

## CANDIDATES

Here are the candidates, each is identified by a unique 'node_id' provided at the very start in [] (e.g., [0]).

{child_node_options}

---

## YOUR EVALUATION TASK
1.  First, identify the essential problem in the query.
2.  Think step by step to reason about why each candidate is relevant or irrelevant (based on the relevance definition). Provide this analysis in the '
reasoning' field.
3.  Rank these passages based on their relevance to the query. Provide your ranking in the 'ranking' field.
4.  Assign a relevance score from 0 to 100 (based on the relevance definition and the ranking). Provide relevances in the 'relevance_scores' field.

---

## OUTPUT FORMAT
You must provide your response as a single, clean JSON object. The JSON should have three keys: 'reasoning', 'ranking', and 'relevance_scores'.

* 'reasoning': This must be a **string**.
* 'ranking': This must be an **array of integers** representing the order of the candidates.
* 'relevance_scores': This must be an **array of arrays** where each inner array contains [node_id, relevance_score]. For example: [[0, 85], [1, 92], [2,
 73]].

---

## YOUR RESPONSE
```

Figure 10: Prompt template used in our experiments for scoring a list of nodes for $\mathcal{L}$.

```
You are an expert in information retrieval and keyword generation. Your task is to analyze a provided list of informational passages and generate a
hierarchically sorted list of search keywords for each passage, strictly adhering to the 5-level rubric below.

## Keyword Generation Rules (5 Levels):

Level 1: 1-2 Word, Core Subject / Domain (Broadest)
Meaning: The absolute fundamental, overarching subject area or discipline.
Characteristics: Only 1 to 2 word, very high-level (e.g., "Technology", "Science", "History")

Level 2: 3-4 Word, General Topic / Sub-domain
Meaning: Narrows Level 1; the specific major topic or branch within the broader field.
Characteristics: Only 3 to 4 words, still general but more focused

Level 3: 4-6 Word, Key Concepts / Main Themes
Meaning: The central ideas, significant concepts, or primary themes directly discussed.
Characteristics: Only 4 to 6 words, core messages, primary subjects, often main sections

Level 4: 7-10 Word, Very Concise Passage Summary
Meaning: A very short, concise summary of what the entire passage is about. This should encapsulate the essential idea or purpose of the passage.
Characteristics: Only 7 to 10 words

Level 5: 11-20 Word, Concise Passage Summary (Most Specific)
Meaning: A concise summary but more descriptive than level 4 of what the entire passage is about. This should encapsulate the main idea or purpose of the
 passage.
Characteristics: A single sentence, 11 to 20 words.

### General Keyword Requirements:

- All keywords must be actionable terms or phrases a user would realistically search.
- Ensure comprehensive coverage of the passage's content across all 5 levels.

## Output Format

Your output must be a single JSON object. This object will contain a top-level key: "passages_keywords". The value associated with this key must be a
JSON array. Each element in this array will be an object with two keys:
"passage_id": An integer that exactly matches the "id" from the corresponding input passage.
"hierarchical_keywords": A JSON array of strings of length 5. Each string represents a hierarchical level (Level 1 at index 0, Level 2 at index 1, and so
 on).

## List of Input Passages:

{desc_list}
```

Figure 11: Prompt template used in our experiments for generating multi-level keywords to be used in top-down tree construction.

```
You are an expert data analyst and taxonomist. Your task is to analyze a list of keywords and their associated counts which indicate how many that
keyword appears in the corpus.

## Goal
- Group the following keywords into **k** semantically coherent and **well-balanced** (i.e. each cluster should aim to contain similar weighted count)
clusters, where k is between [{min_k}, {max_k}]. The primary basis for grouping must be the **topic and meaning** of the keywords.
- Use the provided count as a measure of each keyword's **importance or popularity**. This weight should help you decide which topics are most
significant.
- Try to always maximize the number of clusters but **without** sacrificing the quality of the clustering, **quality of clustering is paramount**.

For every cluster, generate:
* A descriptive `cluster_name`.
* An information-dense `cluster_description` summarizing the core themes.
* A list of all input `keywords` that constitute this cluster or apply to this cluster.

## Input Data
Here is the list of keywords and their importance counts:

{keywords_list_with_count}

## Desired Output Format
Your final output must be a single JSON object, with no other text or explanation. The JSON object must have key: "clusters".

{{
  "clusters": [
    {{"name": "Name of Cluster 1", "description": "A very information dense description of the cluster", "keywords": ["keyword 1", "keyword 2", ...] }},
    {{"name": "Name of Cluster 2", "description": "A very information dense description of the cluster", "keywords": ["keyword 3", "keyword 4", ...] }},
    ...
  ],
}}

---

## Your Response
```

Figure 12: Prompt template used for ClusterLLM to be used in top-down tree construction i.e. clustering a given set of keywords into $[M_{min}, M_{max}]$ clusters.

```
You are an expert AI analyst and summarizer. Your mission is to create a highly informative and "discriminative signpost" for a navigating search agent.
This signpost (a summary) must guide the agent to the correct cluster of nodes to answer a user's query.

You will follow a strict, step-by-step cognitive process. You must analyze the children nodes in a target parent node (the "Positive Set").

Prompt ID: {prompt_id} (ignore, this is just for watermarking purposes).

## INPUTS

### POSITIVE SET: Information about the target parent node to be summarized

{positive_set_descriptions}
---

## YOUR TASK & OUTPUT FORMAT

Your entire output must be a single, valid JSON object. Inside this JSON, you will follow the 3-step thinking process outlined below, populating each
field as instructed.

### JSON Structure and Instructions:

{{
  "detailed_fingerprints": [
    // For EACH children node in the POSITIVE SET (target parent node), extract a structured object of its key, queryable facts.
    {{
      "one_line_summary": "...", // write a very information dense and very concise one-line summary for the information contained in this node
      "key_entities": ["..."], // List a very few key entities which is central to this node
      "genre_or_category": ["..."], // List a few key genre / categories this node can be classified into
      "name": "...", // Name the node
    }}
  ],
  "common_theme": "...", // Reason deeply what are the common themes between the nodes in the POSITIVE SET
  "summary": "...", // Based on step 1 and step 2, write a very information dense description of the target node, **make sure to include all key entities
  **.
}}

---

## Your Response
```

Figure 13: Prompt template for generating bottom-up summaries of a group of nodes.

