# OpenReview forum: "LLM-guided Hierarchical Search for End-to-end Reasoning Intensive Retrieval"
_TMLR — Under review for TMLR_

### Review · Reviewer_t8ou · 2026-06-18

**Summary Of Contributions:**

**Summary**:

The paper proposes **LATTICE**, an LLM-guided hierarchical search framework for reasoning-intensive retrieval, where relevance often depends on semantic relations or multi-step reasoning rather than surface-level similarity. Instead of relying on an embedding retriever to produce a top-k shortlist for an LLM reranker, LATTICE builds an LLM-navigable tree index over the corpus: documents are organized into a hierarchy with LLM-written summaries at internal nodes, and at query time an LLM performs best-first search over the tree. The method introduces top-down LLM-based index construction, cross-branch calibration, and path-aggregated scoring to make tree traversal more robust. Experiments on BRIGHT show that LATTICE is competitive with strong fine-tuned and ensemble retrieval baselines, while LATTICE++ further improves performance by combining LATTICE scores with BM25 and dense retrieval signals. However, the approach has higher latency and cost, making it most suitable for deep, high-value retrieval scenarios rather than ordinary low-cost search.

**Strengths**:

1. The method is well motivated and technically reasonable. LATTICE addresses an important limitation of retrieve-then-rerank pipelines by moving LLM-based reasoning into the retrieval process itself, and its tree-based search framework, together with cross-branch calibration and path-aggregated scoring, provides a coherent solution to reasoning-intensive retrieval.
2. The experiments are extensive and informative. The paper compares LATTICE and LATTICE++ against strong baselines on BRIGHT, includes controlled comparisons with sliding-window reranking, evaluates generalization to traditional IR benchmarks, and provides useful ablations for key design choices.
3. The paper is clearly written and logically organized. The motivation, method, and experimental findings are easy to follow, and the figures help illustrate both the offline index construction and the online search procedure.

**Weaknesses**:

1. LATTICE assumes a hard tree-structured semantic partition, where each document follows a single root-to-leaf path. This design may be brittle for multi-topic or bridging documents, because assigning such documents to only one branch can hide them from queries approaching through another relevant topic. Although the paper mitigates search-time branch-selection errors with beam search, cross-branch calibration, and path aggregation, it does not address construction-time ambiguity or overlapping topic membership.
2. The fixed constraint that each node has at most M children may impose an artificial hierarchy on corpora with highly skewed topic distributions. Dense topics may have to be repeatedly split into deeper subtrees or merged into overly broad intermediate nodes, which could make the tree less faithful to the corpus semantics. This also raises maintainability concerns for dynamic corpora, since inserting new documents may require local re-clustering and summary updates when a node exceeds the branching limit.
3. The failure analysis is currently limited. The paper mainly discusses a dynamic-corpus case where stale tree summaries mislead the search, but it does not systematically attribute errors to index construction, tree traversal, score calibration, or final leaf ranking. A more detailed error taxonomy, especially on static corpora, would help clarify where LATTICE fails and which components are most responsible.

**Audience:**

Yes

**Audience Explanation:**

The paper should be of interest to parts of the TMLR audience, since retrieval-augmented generation are active areas of research, and reasoning-intensive retrieval is an important emerging challenge. The paper also offers a technically novel retrieval paradigm by moving LLM reasoning into the search process itself through a hierarchical index, rather than using the LLM only as a downstream reranker.

**Claims And Evidence:**

Yes

**Claims Explanation:**

The main claims are generally supported by clear and convincing experimental evidence. However, some claims would be better substantiated if the issues discussed in the **Weaknesses** are further addressed.

**Requested Changes:**

1. Some figures (especially Figures 1–3) and tables (especially Table 1) appear to use fonts substantially smaller than the main text, which makes them difficult to read. The authors should improve the readability of these visual elements.
2. The authors should address the concerns discussed in the Weaknesses section.

---

> ### Author Response · Authors · 2026-07-16
>
> We thank the reviewer for the careful reading and the accurate summary of both strengths and limitations. We address the three weaknesses and the requested changes below.
>
> **1. Hard tree partitions and multi-topic / bridging documents.**
>
> We agree this is a real design limitation and now discuss it explicitly in the Limitations section (Appendix A). Three points of context:
>
> - **Construction-time mitigation already present**: for corpora with natural parent-child structure (all StackExchange subsets), the hybrid construction (Sec 2.1) groups passages under their source documents, so the common case of a long multi-topic *document* is handled by its passages living as separate leaves; the single-path constraint binds at the passage level, where topics are far more atomic.
> - **Search-time mitigations**: as the reviewer notes, beam search, cross-branch calibration, and path aggregation reduce the cost of a wrong-branch entry. Coverage loss is modest in practice: LATTICE reaches 74.8% Recall@100 on the StackExchange average, 4.0 points over the fine-tuned ReasonIR-8B retriever (Appendix B.8). If hard partitions were routinely hiding gold documents, we would expect recall to lag flat retrievers rather than lead them.
> - **Path forward**: nothing in the traversal requires a partition. A bridging leaf can be replicated under multiple internal nodes and deduplicated in the prediction set, and extending the ClusterLLM to emit soft or multi-cluster assignments is a natural extension; we have not evaluated it and say so in the revision.
>
> **2. Fixed branching factor $M$ under skewed topic distributions.**
>
> Two clarifications, now reflected in Appendix A. First, the constraint is $M_{\min} \le |C(v)| \le M$ with the LLM choosing the number of clusters per split, and skew is absorbed as *depth* rather than forced merges: a dense topic becomes a deeper subtree, exactly as in hierarchical k-means. The cost of skew is extra traversal depth for queries into dense regions (more LLM calls), not a structural misrepresentation, though we agree intermediate-node fidelity depends on the quality of the LLM's recursive splits, which the depth-adaptive summary levels (Sec 2.1) are designed to support. Second, on maintainability: insertions trigger local re-clustering only when a node exceeds $M$, affecting a single root-to-leaf path, and Appendix B.9 shows a search-guided insertion mechanism that adapts the index with no re-clustering or summary updates and *outperforms* a full static rebuild on TheoremQA-T (49.7 vs. 47.4 nDCG@10). Large-scale continual maintenance (including summary refresh policies) is unevaluated, and the Limitations section says so.
>
> **3. Limited failure analysis / error taxonomy.**
>
> We agree a per-query error-attribution pipeline would strengthen the paper, our existing results do support a coarse quantitative taxonomy, which we have organized into a new Appendix C.3:
>
> - **Coverage vs. ranking**: Recall@100 (74.8% StackExchange average, Appendix B.8) splits errors into gold documents never surfaced (about a quarter; attributable to construction and traversal) vs. surfaced but misranked (the remaining nDCG@10 gap; attributable to scoring and calibration).
> - **Construction errors**: isolated by the matched-traversal ablation (Table 4): 1.8 points on Biology and 12.5 points on TheoremQA-T are lost with a bottom-up index, so construction quality is corpus-dependent and dominates on notation-heavy corpora.
> - **Traversal and scoring errors on static corpora**: isolated by the component ablations. Removing calibration costs 2.4 points on the StackExchange average (22.6 on Biology in the $\ell$ sweep), path aggregation 3.4 points, and explicit reasoning 2.7 points (Table 5, Fig. 5).
> - **Staleness errors**: the dynamic-corpus subsets isolate index-staleness failures (a 9 to 13 point gap to ensembles; qualitative trace in Appendix C.2).
>
> **4. Figure and table fonts.**
>
> We apologize for the readability issues. We have increased the font sizes in Figures 1-3 and Table 1 in the revised PDF.
>
> We hope our response and the revision address the reviewer's concerns, and we are happy to provide further clarifications.

---

### Review · Reviewer_vdZS · 2026-06-19

**Summary Of Contributions:**

This paper proposes LATTICE, an LLM-guided hierarchical search method for reasoning-intensive retrieval. It first constructs a hierarchical index offline via LLM summarization and clustering. During querying, the LLM conducts a best-first search along this index instead of relying on the top-k outputs from dense retrievers. The authors also introduce cross-branch calibration and path-aggregated scoring to mitigate the instability of LLM scoring. This paper presents a clear problem motivation and comprehensive experiments. In particular, the results on the BRIGHT dataset are fairly convincing.

**Additional Comments:**

Overall, I find this an interesting paper with an important research problem and solid experiments. My main reservations lie in its novelty and computational cost. The proposed method essentially achieves a well-combined integration of existing hierarchical retrieval, LLM scoring and tree search, rather than introducing a completely novel mechanism. If the authors refine their statements regarding novelty and add more direct comparative experiments as well as cost analysis, I would lean towards a borderline accept.

**Audience:**

Yes

**Audience Explanation:**

I believe readers focusing on LLM-based retrieval, RAG, reasoning-intensive search and agentic search will find this article intriguing. It addresses a practical real-world problem: when dense retrievers fail to retrieve documents that require complex reasoning to identify, can LLMs be involved in the retrieval process in a more direct manner?

**Broader Impact Concerns:**

None.

**Claims And Evidence:**

Yes

**Claims Explanation:**

Overall, most of the experimental conclusions in the paper are supported. The author compared on BRIGHT with stronger baselines, and also provided same-LLM reranking comparisons and some necessary ablation experiments. These results basically support that LATTICE is effective in reasoning-intensive retrieval.

However, I think the author needs to be more cautious in his description of novelty. The current evidence can show that this combination is effective, but it is not sufficient to prove that it is an entirely new search paradigm.

**Requested Changes:**

1. The author needs to clarify the novelty more clearly. The hierarchical index, LLM clustering, LLM listwise scoring, best-first tree search and score calibration adopted in the current method all draw from existing work. It is recommended to make a clearer distinction between existing components and the truly novel designs proposed in this paper.
2. Hope to add more comparable ablation studies. Examples include RAPTOR-style tree combined with LLM traversal, LLM-constructed tree paired with simple greedy search, and a reranking baseline under the same LLM budget. This can clarify whether performance improvements stem from tree construction, search algorithms, or increased LLM computation volume.
3. A more thorough discussion of costs and applicable scenarios is required. Both the offline indexing and online search of LATTICE rely on a large number of LLM calls, resulting in substantial actual latency and update costs.
4. Its performance under dynamic corpus settings is subpar, which merits further analysis. Static hierarchical indexes may encounter similar issues in scenarios involving document filtering, updates, or personalization.

---

> ### Author Response · Authors · 2026-07-16
>
> We thank the reviewer for the thoughtful review and for stating a clear path to acceptance. We address the four requested changes below.
>
> **1. Clarify novelty: distinguish existing components from novel designs.**
>
> We agree and have revised the framing (jointly requested by Reviewer tRsK). The revision (a) removes the "new search paradigm" claims and describes LLM-guided hierarchical search as an alternative direction with clear antecedents (RAPTOR, TopicGPT, listwise LLM ranking, Tree-of-Thoughts style tree search, and the corpus-graph line GAR/QUAM/SlideGAR), and (b) states explicitly what this paper adds: the top-down recursive LLM-supervised construction with multi-level summaries as a context-adaptive interface; the cross-branch calibration and path-aggregated latent scoring that make off-the-shelf listwise scores usable as a tree-search value function; and controlled evidence that the integrated system matches fine-tuned ensemble state of the art. We believe this accurately reflects the evidence, and we thank the reviewer for pushing us to state it precisely.
>
> **2. More comparable ablation studies.**
>
> We would like to respectfully point out that all three requested ablations are already in the submission; we apologize if they were hard to find and have added signposting in Sec 3.4. Concretely:
>
> - **"RAPTOR-style tree combined with LLM traversal"**: this is exactly Table 4. Both rows use the *identical* LATTICE traversal (same LLM, prompts, calibration, budget); only the index differs: bottom-up embed-and-cluster (RAPTOR-style) vs. top-down LLM-guided, under matched structural constraints. Top-down wins by +1.8 nDCG@10 on Biology and +12.5 on TheoremQA-T. This isolates tree construction from the search algorithm.
> - **"LLM-constructed tree paired with simple greedy search"**: the components separating our search from greedy descent are each ablated on the LLM-built tree. Trusting raw local scores (no calibration, $\ell = 0$, the core of greedy traversal) collapses Biology from 69.5 to 46.9 (Fig. 5 left); removing path aggregation ($\alpha = 0$, purely local decisions) costs 3.4 nDCG@10 on the StackExchange average (Table 5); a single-path frontier ($B = 1$) is in the budget-matched sweep (Fig. 5 right). The gains come from the robust-search mechanisms, not from the tree alone.
> - **"A reranking baseline under the same LLM budget"**: this is the controlled comparison of Sec 3.2 / Fig. 4: a sliding-window reranker over a strong dense shortlist with the *same* LLM and *same* scoring prompt, plotted as token-budget-matched curves. At matched budgets past the crossover, LATTICE is strictly better (+3.8 to +5.3 on 3 of 4 subsets), so the improvement does not stem from more LLM computation alone.
>
> **3. A more thorough discussion of costs and applicable scenarios.**
>
> We have consolidated and extended the cost discussion (Sec 3.2, Appendix B.7):
>
> - **Online**: LATTICE costs \\$0.06 to \\$0.38 per query depending on the LLM (Table 2). We now also report estimated costs for the ensemble baselines: DIVER v3 at about \\$0.06 per query and the full BGE-Reasoner-0928 ensemble at about \\$0.71 per query at open-weight rates. LATTICE is therefore cost-comparable to, or cheaper than, the systems it matches on quality, though at higher latency (tens of seconds).
> - **Offline**: index construction is a one-time cost of about 6 hours for a 100K-document corpus, amortized over all queries; incremental updates do not require a rebuild (Appendix B.9).
> - **Applicability**: we state directly (Intro, Sec 3.3, Conclusion) that LATTICE targets quality-over-latency workloads (deep research, legal and technical QA), and that on traditional workloads where cheap retrievers do not structurally fail, it does not add enough value to justify its cost (Sec 3.3).
>
> **4. Dynamic corpus performance.**
>
> We agree this is the method's current limitation (Sec 3.2; qualitative failure trace in Appendix C.2). Two clarifications. First, BRIGHT's dynamic subsets are an unusually adversarial setting: more than 10K documents are removed *per query*, which differs from typical production churn (incremental insertion and deletion). Second, for the incremental setting we provide a quantitative mitigation: the search-and-insert mechanism of Appendix B.9 inserts new documents via the search procedure itself, without rebuilding, and the resulting tree *outperforms* the static full rebuild on TheoremQA-T (49.7 vs. 47.4 nDCG@10; Recall@100 improves from 73.9 to 81.1). Per-query exclusion at BRIGHT's scale remains open, and the revised Limitations section now says so explicitly; LATTICE++ offers a practical stopgap by fusing exclusion-aware cheap retrieval.
>
> We hope the revised framing, the ablation signposting, and the extended cost discussion address the reviewer's reservations, and we thank them again for the constructive path they laid out. We are happy to discuss further.

---

### Review · Reviewer_tRsK · 2026-07-01

**Summary Of Contributions:**

This paper proposes LATTICE, an LLM-guided hierarchical search framework for reasoning-intensive retrieval. Instead of relying on a embedding -based retriever at query time, LATTICE constructs an offline semantic tree which can be navigated directly by a search LLM. The main technical components include top down LLM-guided tree construction, cross-branch calibration for LLM scoring that depends on the candidate set, and path-relevance aggregation for best-first traversal. Experiments on the BRIGHT dataset show that LATTICE performs comparably to strong fine-tuned ensemble baselines, while LATTICE++ further improved its performance by combining LATTICE with BM25 and dense retrieval signals.

**Audience:**

Yes

**Audience Explanation:**

Yes. The paper addresses a timely problem in reasoning-intensive retrieval and proposes an interesting alternative to the dominant retrieve-then-rerank paradigm. The idea of letting an LLM directly navigate a semantic hierarchy should be of interest to researchers working on IR, RAG, LLM agents, and search systems.

**Broader Impact Concerns:**

I do not see broader impact concerns that would require adding a Broader Impact Statement. The paper is mainly methodology, and the risks appear to be standard retrieval-system risks rather than unusually severe societal harms.

**Claims And Evidence:**

Yes

**Claims Explanation:**

Yes, most of the main empirical findings are supported. The paper includes a rigorous comparison with sliding-window reordering using LLMs, ablation experiments on calibration and path dependence, results from open-weight LLMs, and additional BEIR experiments. However, clarification is needed before the claims are fully convincing. Although the latent-score calibration model is specified in Eq. 2–3 and implementation details are given in the appendix, the paper does not clearly discuss identifiability and stability of the fitted latent scores. Since $s_v^i \approx a \hat{s}_v + b_i$, the scale and shift of $\hat{s}_v$, $a$, and $b_i$ may be underdetermined without additional constraints or normalization. This is important because $\hat{s}_v$ is later used directly in the path-relevance update. It would be useful to clarify how the estimator is constrained, normalized, or stabilized, especially for nodes that appear in only one slate.

**Requested Changes:**

1. (Important) Clarify the identifiability and stability of the latent-score estimator. Please state explicitly what constraint or normalization pins down the scale/shift degeneracy in
$s_v^i \approx a\hat{s}_v + b_i$, for example by fixing $a$, centering $b_i$, or normalizing $\hat{s}_v$.
Please also clarify how $s_v$ is estimated or initialized for nodes that appear in only one slate,
where the latent score is weakly constrained by cross-slate comparisons. A short sentence in Sec. 2.2 plus the constraint in Appendix B.1.2 would be enough.

2. (Important) Position the contribution more carefully against prior work. The core building blocks — semantic hierarchical indexing (RAPTOR), LLM-based clustering (TopicGPT), LLM listwise scoring, and best-first tree search (Tree-of-Thoughts) — are each established. The evidence supports that the integration is effective, not that any single component is novel, so the "new paradigm" framing should be tempered with a clearer statement of what is genuinely new.

3. (Minor) Surface the base-vs-ensemble decomposition more clearly in the main text.
The per-component decomposition in Appendix B.5 / Table 7 is useful, but the main text should more explicitly distinguish the contribution of base LATTICE from the gains due to LATTICE++ ensemble fusion. In particular, when discussing the 49.1 nDCG@10 result, it would help to briefly remind readers that this number comes from fusing LATTICE with BM25 and dense retrieval signals, whereas the base hierarchical-search method reaches 46.7. This would make the interpretation of the headline result clearer.

---

> ### Author Response · Authors · 2026-07-16
>
> We thank the reviewer for the thorough review and the concrete, actionable requested changes. All three are implemented in the revised PDF.
>
> **1. Identifiability and stability of the latent-score estimator.**
>
> The reviewer is correct, and we apologize for not stating the constraints explicitly. As the reviewer notes, Eq. 2 leaves the scale and shift of $(a, \{\hat{s}_v\}, \{b^i\})$ underdetermined: the reparameterization $(a, \hat{s}_v, b^i) \to (a/c, \ c\hat{s}_v + d, \ b^i - (a/c)d)$ preserves all predictions. Following the reviewer's suggestion, we added a short statement to Section 2.2 and the full specification to Appendix B.1.2:
>
> - **Pinning the degeneracy.** All parameters are initialized at neutral values ($a=1$, $b^i=0$, and $\hat{s}_v$ near 0) and Eq. 3 is minimized by Adam with a small $\ell_2$ penalty (weight decay) on all parameters. The ridge penalty is strictly convex along the degenerate directions and selects the minimum-norm solution in the equivalence class, which is functionally the same as the constraints the reviewer suggests (fixing $a=1$ or centering $b^i$). After fitting, latent scores are max-normalized, $\hat{s}_v \leftarrow a\hat{s}_v / \max_u a\hat{s}_u$, anchoring the top node to 1 so that the latent scores live on the same $[0,1]$ scale as the raw scores and the root's path relevance, the scale at which Eq. 4 consumes them. The induced ranking of nodes is invariant to the affine ambiguity in any case; the normalization pins the one downstream use (EMA mixing in Eq. 4) where absolute scale matters.
> - **Nodes appearing in only one slate.** By construction (Section 2.2, "Slate construction"), every slate contains calibration anchors already scored in earlier slates: the highest-scoring cross-branch sibling for internal slates, and $\ell = 10$ top-scoring leaves from the prediction set for leaf slates. These recurring anchors keep the node-slate incidence graph connected and pin each slate's bias $b^i$. A node observed in a single slate is then identified by its raw score relative to the anchored slate, without requiring multiple observations of the node itself. The ridge penalty keeps the problem well-posed even in the corner case of an anchor-free slate. The calibration sweep (Fig. 5 left) is consistent with this analysis: with no anchors ($\ell = 0$) quality degrades severely (46.9 vs. 69.5 nDCG@10 on Biology) and stabilizes by $\ell$ of 5 to 10.
>
> **2. Position the contribution more carefully against prior work.**
>
> We agree with this characterization and have revised the framing accordingly. The building blocks (semantic hierarchical indexing from RAPTOR, LLM-based clustering from TopicGPT, listwise LLM scoring, and best-first tree search from Tree-of-Thoughts) are each established, and our evidence supports that the integration is effective, not that any single component is new. In the revision:
>
> - The abstract and introduction no longer describe the work as a "new search paradigm". The introduction now states directly: the building blocks are established; our contribution is showing why the naive combination fails, the mechanisms that fix it, and that the result is competitive with the strongest fine-tuned systems.
> - What is new, stated explicitly: (i) the top-down *recursive* LLM-supervised construction using multi-level summaries as a context-budget-adaptive interface, extending TopicGPT-style flat topic discovery to recursive hierarchy construction at the scale of $10^4$ to $10^6$ documents (shown to beat bottom-up construction under matched structure, Table 4); (ii) cross-branch calibration anchors with slate-debiased latent scoring, addressing a failure mode (slate dependence) that flat rerankers sidestep with sliding windows but that is fatal for tree search (Fig. 5 left); and (iii) the controlled same-LLM, same-prompt comparison isolating the search algorithm's contribution (Fig. 4).
>
> **3. Surface the base-vs-ensemble decomposition in the main text.**
>
> Done. Where the 49.1 result is reported (Sec 3.2), the text now states that 49.1 fuses LATTICE with BM25 and BGE-Reasoner-Embed, that the base hierarchical-search method alone reaches 46.7, and points to the per-component decomposition in Appendix B.5 / Table 7. The same paragraph notes where the ensemble's gains concentrate (the dynamic-corpus subsets).
>
> We hope our response and the revision address the reviewer's concerns, and we are happy to provide further clarifications.

---

### Review · Reviewer_d8N5 · 2026-07-04

**Summary Of Contributions:**

The paper proposes LATTICE, an LLM-guided hierarchical search framework that removes the embedding model from the query-time loop. It builds an LLM-navigable semantic tree offline and has an LLM walk the tree via best-first frontier search  at query time. The framework includes a top-down, LLM-supervised tree construction over multi-level document summaries, and a traversal algorithm that mitigates the noise of listwise LLM scoring through cross-branch calibration nodes. On the BRIGHT benchmark, LATTICE shows promising results.

**Audience:**

Yes

**Audience Explanation:**

This work would be of interest to the IR, RAG, and LLM-agent communities, particularly given its well-motivated grounding in the theoretical limits of dense embeddings on reasoning-intensive queries. The traversal techniques are reusable beyond this specific system and useful to anyone employing LLMs as rankers.

**Claims And Evidence:**

Yes

**Claims Explanation:**

The paper's claims are well supported and the ablations provides convincing evidence. The calibration sweep shows a large collapse in quality when cross-branch calibration is removed, the budget-matched sweep establishes that depth beats breadth, and the component ablation isolates each mechanism and demonstrates its individual contribution. Scalability is credibly demonstrated on a million-document corpus, and the open-weight cost analysis supports the LLM-agnostic claim.

**Requested Changes:**

1. It would be helpful to state the identifiability and normalization constraints for the latent-score model, and to clarify how the latent score is mapped back into the score range used for path relevance.

2. Estimating the inference cost of the ensemble baselines would make the comparison more complete.

3. Please clarify whether LATTICE operates over the raw or the rewritten query.

---

> ### Author Response · Authors · 2026-07-16
>
> We thank the reviewer for the positive assessment and the precise requested changes. We address each point below; all changes are in the revised PDF.
>
> **1. Identifiability and normalization constraints for the latent-score model, and how the latent score maps back to the path-relevance score range.**
>
> We apologize for leaving these constraints implicit in the submission. The reviewer is right that Eq. 2 as written is underdetermined: for any $c \neq 0$ and $d$, the reparameterization $(a, \hat{s}_v, b^i) \to (a/c, \ c\hat{s}_v + d, \ b^i - (a/c)d)$ leaves all predictions unchanged. Our implementation resolves this in two steps, now stated in Section 2.2 with full details in Appendix B.1.2:
>
> - **Fitting.** All parameters are initialized at neutral values ($a=1$, $b^i=0$, and $\hat{s}_v$ near 0) and Eq. 3 is minimized with Adam under a small $\ell_2$ penalty (weight decay) on all parameters. The ridge term is strictly convex along the degenerate directions and selects the minimum-norm solution, which plays the same role as fixing $a=1$ or centering the biases.
> - **Mapping back.** After each refit, latent scores are rescaled as $\hat{s}_v \leftarrow a\hat{s}_v / \max_u a\hat{s}_u$, anchoring the top-scoring node to 1. This places the latent scores on the same $[0,1]$ scale as the raw LLM scores and the root's path relevance, which is exactly the scale at which Eq. 4 mixes $\hat{s}_v$ with ancestral evidence.
>
> We also note that the affine ambiguity is shared across all nodes and cannot change their ranking; the max-normalization pins the one downstream quantity where absolute scale matters (the EMA mixing in Eq. 4).
>
> **2. Inference cost of the ensemble baselines.**
>
> We agree this makes the comparison more complete and have added it (Sec 3.2, Appendix B.7). Accounting the published pipelines at open-weight (DeepInfra) per-token rates:
>
> - **DIVER v3** (14B pointwise + 32B listwise reranking over the top-100 shortlist): roughly 243K input and 44K output tokens per query, which comes to about \$0.06 per query.
> - **BGE-Reasoner-0928 full ensemble** (8B, 14B, and 32B rerankers over six retriever-by-size combinations, roughly 3,500 reranker calls per query): roughly 3.0M input and 0.9M output tokens per query, which comes to about \$0.71 per query.
>
> For reference, LATTICE runs at \\$0.38 per query (Gemini-3-Flash), \\$0.11 (Qwen3.5-27B), and \\$0.06 (Gemini-3.1-Flash-Lite) (Table 2). LATTICE is therefore cost-comparable to DIVER v3 at the Flash-Lite/Qwen operating points and substantially cheaper than the full BGE-Reasoner ensemble. We note in the paper that baseline costs are estimates from published pipeline descriptions, not measurements of the authors' serving stacks.

---

### Author Response · Authors · 2026-07-16
**General response**

We sincerely thank all four reviewers for their constructive and careful reviews. We are encouraged that all reviewers found the claims well supported and the findings of interest to the TMLR audience. We have uploaded a revised PDF that implements every requested change. In summary:

1. **Identifiability of the latent-score estimator** (d8N5, tRsK): Section 2.2 now states the affine degeneracy of Eq. 2, the constraint that resolves it, and the normalization that maps latent scores back to the $[0,1]$ scale used by the path-relevance update. Appendix B.1.2 gives the full estimator specification, including how nodes appearing in a single slate are identified.
2. **Novelty framing** (tRsK, vdZS): We have tempered the "new search paradigm" language throughout and added an explicit statement separating established building blocks from what this paper contributes.
3. **Base vs. ensemble** (tRsK): The main text now explicitly distinguishes base LATTICE (46.7) from the LATTICE++ fusion (49.1) where the headline number is reported.
4. **Query input** (d8N5): Clarified that LATTICE operates on the raw query; rewritten queries appear only in the cheap components of LATTICE++ and in baselines.
5. **Costs** (d8N5, vdZS): Added per-query cost estimates for the ensemble baselines and a consolidated cost/applicability discussion.
6. **Figure/table readability** (t8ou): Enlarged fonts in Figures 1–3 and Table 1.
7. **Limitations & failure analysis** (t8ou, vdZS): Expanded the limitations discussion (hard tree partitions, branching-factor constraint) and added an error-taxonomy section (Appendix C.3) organizing our quantitative evidence by failure stage.

We respond to each reviewer individually below and are happy to discuss further.